# A novel mouse model for *LAMA2*-related muscular dystrophy with analysis of molecular pathogenesis and clinical phenotype

Dandan Tan[1,2], Yidan Liu[1], Huaxia Luo[1], Qiang Shen[3], Xingbo Long[4], Luzheng Xu[5], Jieyu Liu[1], Nanbert A Zhong[6]*, Hong Zhang[3]*, Hui Xiong[1,7]*

[1]Department of Pediatrics, Peking University First Hospital, Beijing, China; [2]Department of Neurology, the First Affiliated Hospital, Jiangxi Medical College, Nanchang University, Nanchang, China; [3]Institute of Cardiovascular Sciences and Key Laboratory of Molecular Cardiovascular Sciences, Peking University Health Science Center, Beijing, China; [4]Department of Urology, Sun Yat-sen University Cancer Center, Guangzhou, China; [5]Medical and Health Analysis Center, Peking University, Beijing, China; [6]New York State Institute for Basic Research in Developmental Disabilities, Valhalla, United States; [7]Beijing Key Laboratory of Molecular Diagnosis and Study on Pediatric Genetic Diseases, Beijing, China

*For correspondence:
nanbert.zhong@opwdd.ny.gov
(NAZ);
zhanghong@bjmu.edu.cn (HZ);
xh_bjbj@163.com (HX)

Competing interest: The authors declare that no competing interests exist.

## eLife Assessment

This **useful** manuscript reports on a new mouse model for LAMA2-MD, a rare but very severe congenital muscular dystrophy. The knockout mice were generated by removing exon3 in the Lama2 gene, which results in a frameshift in exon4 and a premature stop codon. These animals lack any laminin-alpha2 protein and confirm results from previous Lama2 knockout models. Additionally, this study includes weak transcriptomics data that might be a good resource for the field. However, experimental evidence, methods, and data analyses supporting the main claims of the manuscript are **incomplete**.

**Abstract** Our understanding of the molecular pathogenesis of *LAMA2*-related muscular dystrophy (*LAMA2*-MD) requires improving. Here, we report the phenotype, neuropathology, and transcriptomics data (scRNA-seq and bulk RNA-seq) of a new *Lama2* knockout mouse (dy$^H$/dy$^H$) which was created based on the human *LAMA2*-MD mutation hotspot region using CRISPR-Cas9. The dy$^H$/dy$^H$ mice presented a severe phenotype with muscular dystrophy. Mouse brain scRNA-seq showed that *Lama2* gene was expressed predominantly and specifically in vascular and leptomeningeal fibroblasts and vascular smooth muscle cells, and weakly in astrocytes in wild-type mouse. Laminin α2 expression on the cortical surface was observed with immunofluorescence. In dy$^H$/dy$^H$, *Lama2* expression was decreased in those cell types, which might be associated with the disruption of gliovascular basal lamina assembly. Additionally, transcriptomic investigation of muscles showed 2020 differentially expressed genes, mainly associated with the impaired muscle cytoskeleton and development. In summary, this study provided potentially useful information for understanding the molecular pathogenesis of *LAMA2*-MD.

## Introduction

*LAMA2*-related muscular dystrophy (*LAMA2*-MD) is an autosomal recessive disorder caused by pathogenic variants in the *LAMA2* gene (*Zambon et al., 2020*). The clinical features of *LAMA2*-MD can be divided into two subgroups: (1) severe, early-onset *LAMA2*-related congenital muscular dystrophy (*LAMA2*-CMD, OMIM 607855), and (2) mild, late-onset autosomal recessive limb-girdle muscular dystrophy-23 (LGMDR23, OMIM 618138) (*Tan et al., 2021*). *LAMA2*-MD has an estimated prevalence of 4 in 500,000 children (*Nguyen et al., 2019*), and accounts for 36–48% of patients with congenital muscular dystrophies (CMDs) (*Sframeli et al., 2017*; *Abdel Aleem et al., 2020*; *Ge et al., 2019*). Muscle weakness with dystrophic muscle pathology and abnormal brain white matter hyperintensities on T2-weighted magnetic resonance imaging (MRI) are the common characteristics of *LAMA2*-MD patients. Some patients present with brain dysfunctions such as seizures, cognitive delay, and cerebral malformation such as occipital pachygyria (*Tan et al., 2021*).

The *LAMA2* gene (OMIM 156225), located on chromosome 6q22.33, encodes the laminin α2 chain which was predominantly localized to the basal lamina of myofibers in skeletal muscle, Schwann cells in peripheral nerve, and cerebral blood vessels detected by immunofluorescence and immunoelectron microscopy (*Leivo and Engvall, 1988*; *Villanova et al., 1997*; *Malandrini et al., 1997*). Laminin α2 connects with laminin β1 and γ1 chains to form a heterotrimeric protein laminin-α2β1γ1 (LM-211). LM-211 anchors to two major groups of receptors on the cell surface, α-dystroglycan and integrins (α1β1, α2β1, α6β1, α7β1), to form dystroglycan-matrix scaffolds and regulate signal transductions (*Hohenester, 2019*; *Durbeej, 2010*; *Aumailley, 2021*). It was reported that the deficiency of laminin α2 in astrocytes and pericytes was associated with a defective blood-brain barrier (BBB) in the dy$^{3K}$/dy$^{3K}$ mice (*Menezes et al., 2014*). The defective BBB presented with altered integrity and composition of the endothelial basal lamina, reduced pericyte coverage, and hypertrophic astrocytic endfeet lacking appropriately polarized aquaporin-4 channels. It has been suspected that laminin α2 deficiency is responsible for disruption of the basement membrane assembly (*Yurchenco et al., 2018*), leading to defects in the gliovascular basal lamina of the BBB (*Gawlik and Durbeej, 2020*; *Menezes et al., 2014*; *Arreguin and Colognato, 2020*), and the muscle damage.

Mouse models are the most frequently used to investigate the pathogenic mechanism for muscular dystrophies. The most common mouse models for *LAMA2*-MD are the dy/dy, dy$^{3k}$/dy$^{3k}$, dy$^{w}$/dy$^{w}$, and dy$^{2J}$/dy$^{2J}$ mice (*Xu et al., 1994*; *Michelson et al., 1995*; *Miyagoe et al., 1997*; *Kuang et al., 1998*; *Sunada et al., 1995*). Among them, the dy/dy, dy$^{3k}$/dy$^{3k}$, dy$^{w}$/dy$^{w}$ mice present severe muscular dystrophy, and dy$^{2J}$/dy$^{2J}$ mice show mild muscular dystrophy and peripheral neuropathy (*Gawlik and Durbeej, 2020*). The mutation of the dy/dy mice has still been unclear (*Xu et al., 1994*; *Michelson et al., 1995*). The dy$^{3k}$/dy$^{3k}$ mice were generated by inserting a reverse Neo element in the 3' end of exon 4 of *Lama2* gene in 1997 (*Miyagoe et al., 1997*), and the dy$^{w}$/dy$^{w}$ mice were created with an insertion of lacZ-neo in the exon 1 of *Lama2* gene in 1998 (*Kuang et al., 1998*). The dy$^{2J}$/dy$^{2J}$ mice were generated in 1970 by a spontaneous splice donor site mutation which resulted in a predominant transcript with a 171 base in-frame deletion, leading to the expression of a truncated laminin α2 with a 57 amino acid deletion (residues 34–90) and a substitution of Gln91Glu (*Sunada et al., 1995*). However, they were established in the pre-gene therapy era, leaving a trace of engineering, such as bacterial elements in the *Lama2* gene locus, thus unsuitable for testing various gene therapy strategies. Moreover, insufficient transcriptomic data of the muscle and brain of *LAMA2*-CMD mouse models limits the understanding of disease hallmarks. Therefore, there is a need to create new appropriate mouse models for *LAMA2*-CMD based on human high-frequency mutated region using the latest gene editing technology, such as clustered regularly interspaced short palindromic repeats (CRISPR)-Cas9.

Our previous studies have suggested that exons 3–4 region of the *LAMA2* gene is the most frequent mutation region in patients with *LAMA2*-MD (*Tan et al., 2021*; *Ge et al., 2018*). Building on this finding, we utilized CRISPR-Cas9 gene editing technology to specifically delete the exon 3 of *Lama2* gene, creating a novel *Lama2* knockout (KO) mouse model for *LAMA2*-MD, named 'dy$^{H}$/dy$^{H}$'. Through the analysis of phenotype, the muscle pathology, and the expression of laminin α2 protein, we confirmed that dy$^{H}$/dy$^{H}$ mice presented a phenotype with severe muscular dystrophy. Then, we conducted single-cell RNA sequencing (scRNA-seq), RNA sequencing, western blot, and immunofluorescence to provide more useful data and information for further investigating the molecular pathogenetic mechanisms of muscular dystrophy and brain dysfunction for *LAMA2*-CMD.

# Results

## dy$^H$/dy$^H$ mice: a novel mouse model for *LAMA2*-MD

We have created a new mouse model dy$^H$/dy$^H$ for human *LAMA2*-MD that closely simulates human neuropathology and clinical phenotype. This was generated by targeted deletion of *Lama2* gene exon 3 with CRISPR-Cas9 gene editing technology (*Figure 1*; *Barazesh et al., 2021*). To assess the phenotype of these *Lama2* KO mice, we recorded the general appearance, body weight, life span, and motor function of wild-type (WT), heterozygote (Het, dy$^H$/+), and homozygous dy$^H$/dy$^H$ mice (*Figure 2*). The dy$^H$/dy$^H$ mice were smaller with lower weight when compared with the WT and Het mice (*Figure 2A*, *Figure 2—figure supplement 1*). The weight gain of dy$^H$/dy$^H$ mice was delayed at postnatal days 7 (P7) and worsened through P10 to P24 (*Figure 2A*). Kaplan–Meier survival analysis revealed that the median survival of dy$^H$/dy$^H$ mice was 21 days (ranged from 12 to 35 days) (*Figure 2A*), while WT and Het mice were in good health throughout the observation period of one year. The dy$^H$/dy$^H$ mice were less active than the WT and Het mice. At P7, the dy$^H$/dy$^H$ mice experienced difficulties in maintaining an upright stance and righting from supine. By P14, these mice were not only less active and smaller than their WT counterparts, but also displayed significant muscle weakness. This weakness progressed rapidly, reaching a debilitating state at age of 3–4 weeks. The average four-limb grip strength of the dy$^H$/dy$^H$ mice presented slow improvement over the course of 21 days post birth, then followed by a subsequent decline (*Figure 2A*). Meanwhile, the four-limb grip strength of the dy$^H$/dy$^H$ mice remained consistently lower than that of the WT and Het mice at all time points (P10, P14, P17, P21, and P24) (*Figure 2A*). During the period from P18 to P24, the dy$^H$/dy$^H$ mice demonstrated inferior performance on the treadmill with a significantly higher number of electric shocks compared to their WT counterparts at time points of P18, P19, P21, P22, P23, and P24 (*Figure 2A*). These observations collectively indicated a severe phenotype in the dy$^H$/dy$^H$ mice, characterized by diminished body weight, severe muscle weakness, and a shortened lifespan.

To further investigate the muscle pathology of the dy$^H$/dy$^H$ mice, we measured the serum creatine kinase (CK) levels, conducted muscle magnetic resonance imaging (MRI) studies, and performed muscle histopathological analyses.

1. Serum CK: The mean serum CK levels of the WT, heterozygous (Het) and dy$^H$/dy$^H$ mice were 87.68±11.92 ng/mL, 82.80±4.57 ng/mL, and 769.67±395.54 ng/mL at P14, and 77.85±14.95 ng/mL, 69.11±6.42 ng/mL, and 786.29±415.58 ng/mL at P21, respectively. The result showed that the mean serum CK levels were significantly elevated in the dy$^H$/dy$^H$ mice when compared to the WT mice at both postnatal days 14 (P14) (p=0.004) and 21 (P21) (p=0.009). There was no discernible difference in serum CK levels between the WT and heterozygous (Het) mice (*Figure 2A*).

2. Muscle MRI: The mean relative muscle areas on T1W of pelvic and hindlimb muscle MRI of the WT and dy$^H$/dy$^H$ mice were 3.58±0.96 and 1.84±0.73 at P14, and 8.52±4.98 and 3.77±0.63 at P21, respectively, indicating substantially smaller muscle volumes in the dy$^H$/dy$^H$ mice at both P14 (p<0.001) and P21 (p<0.001). Additionally, on T2-weighted MRI, the mean relative hyperintense levels of the WT and dy$^H$/dy$^H$ mice were 96.17 and 136.85 at P14, and 96.99 and 135.32 at P21, respectively, indicating significantly increased hyperintensity in the dy$^H$/dy$^H$ muscles at both P14 (p<0.001) and P21 (p<0.001) (*Figure 2B*).

3. Muscle histopathology: Compared to that of age-matched WT mice, biceps femoris muscle of the dy$^H$/dy$^H$ mice revealed increased variation in fiber size at P7, severe dystrophic changes, including muscle fiber degeneration, necrosis, and regeneration at P14 (*Figure 3*, *Figure 3—figure supplement 1*). The significantly increased connective tissue infiltration and inflammation were found in the biceps femoris muscle of dy$^H$/dy$^H$ muscles from P7 (*Figure 3*, *Figure 3—figure supplement 1*). Similar dystrophic features were observed in other muscles including quadriceps femoris, gastrocnemius, triceps brachii, diaphragm, and tongue (*Figure 4A*, *Figure 4—figure supplement 1*). Notably, heart and intestinal smooth muscles were spared from dystrophic changes (*Figure 4A*, *Figure 4—figure supplement 1*). Western blot analysis revealed that the 300 kDa laminin α2 protein was absent in the dy$^H$/dy$^H$ muscle samples, but it was normally expressed in the WT muscle samples (*Figure 4B*), strongly indicating a complete loss of laminin α2 protein in the dy$^H$/dy$^H$ mice. Immunofluorescence staining further confirmed the deficiency of laminin α2 protein in muscle tissue of the dy$^H$/dy$^H$ mice (*Figure 4C*).

These findings collectively indicated that the phenotype of dy$^H$/dy$^H$ mice was the severe muscular dystrophy, and established a novel mouse model for human *LAMA2*-CM.

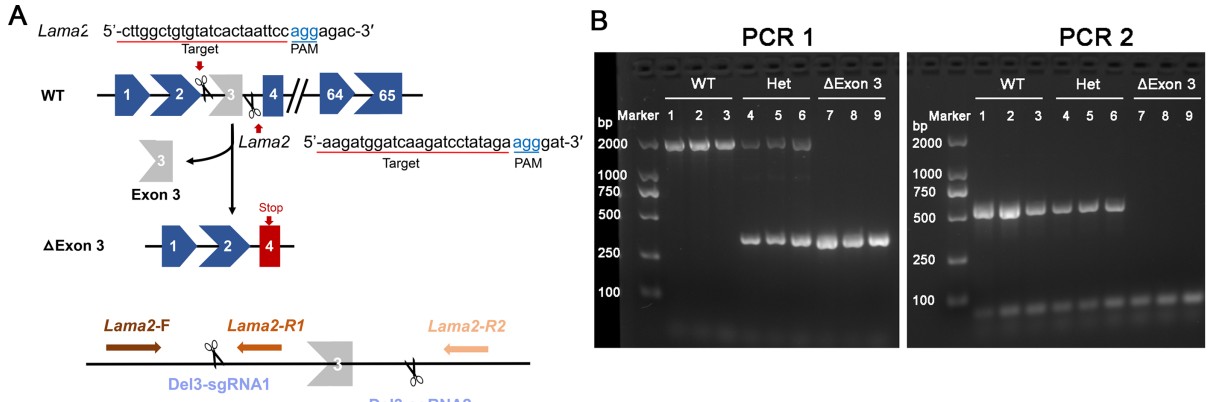

cagcatccttactgaacccaggctccctttgaatcccacaatagactttcttcaaatctgtctggatttggtttgtgtgggaaagcttgtgt
ttggttgttacattgccagccatgtaacagtcctctcctggctaataaatgttgactagattggtctcctggaattagtgatacacagccaa
gcaaatactgcataaggggcttggggaatcttccaatatagttttatatcatgatagaaaggtaacaagtttaaaggtttgttaggaaaaaac
ctcatagacagaaaatgctatcattttatagtaaagtcacatatataccggaatgcccaagcctctgcaatgaagaaaatatatgaataaaa
aatattcaagcacctttccaacaggagagttattgtttatcagatgatcattatgctatttggattcctgtactattcttttgcacttagtt
tctggaggtatgtctaaggctggtggtcagtgtttagtaccttttttgtagcacagtcatgatgacatcatttcctgcatgttgtgtagaat
atgaaaacagaggtggttcgatgtctaatacttgttatccacaggcttgtgatacatcattttgtttttgccttttctttgaagagacatgta
taaactcatcatctgatctttctcctccttctcatgctgctgccaattccatagAGAGGCACCCGATTACGAATGCTATTGATGGCAAGA
ACACATGGTGGCAGAGTCCCAGTATCAAGAATGGAGTGGAATACCATTATGTGACAATTACTCTGGATTTACAGCAGgtaaagtatacctc
ctgtcactttccacttctgaaattctaccttacatctgctatttgtctagttagcaatttttagatgaagggtgaatcgtaatgattcttat
cactattattttcttgatattgtgacaaagggtcctgtctgtttcacagataaccatttattgatgtttggcacttatttctaaagcacaac
tcaaattgacctatgtgggtggttcagaaatgaattagaagctaagtgtgagttacaaagttgggagcgatgctgaaaatatggatcaacaa
aattttatgggaagggaacacaacagcacagtttaaaacttgtaaaacaaaactttgataccactctgctgatagtgtggctcttaaaagcc
acttggatagcaaggctcttgagcaaattatttcttgttcctgagagttgagcaagctgtgacagttgattttcaaattaaacttggaccac
taaaatatgtaaaatttttgttagaagtcaaagattgtttgcataaacttgcatgccttctgttcttaactttttcattttattcacaata
ttcacatcaggctaagttcaaaaaccaggactcatcctcattcccaaattatttagggttacagacggatttgtgatgtaatttgaaaagac
acttatatatttttttctcggaataataagatagtgtcatggaccagagataatgacaagaaatattgaactttgatagtttattattatgtct
attttactttcatactttgggtcgttagagaaaaaaaatcatccagctgcccctcctatgtcagtgtggagataaactaatcagaaaataa
ggggacactgaagatgttctctaacagactacaccatgaggtcatagttatcataggatgccacgggtgagaaaataacagctaacagaat
agatagaaactagacccaagaattaagcatcccttctataggatcttgatccatcttggtaatggcagcagcacacatttctaagttacaaa
tggctgctctcggaatgaggaagttggtaggtctcaaaaaagagcctcctttatccctccatggactccttggcacctaagtaactgggtag
cagcagggtgcaaacgctactgctaaatgaccagctaacctgggtttattcctattagcagggtgtcagccttgaggttagttcttaattcc

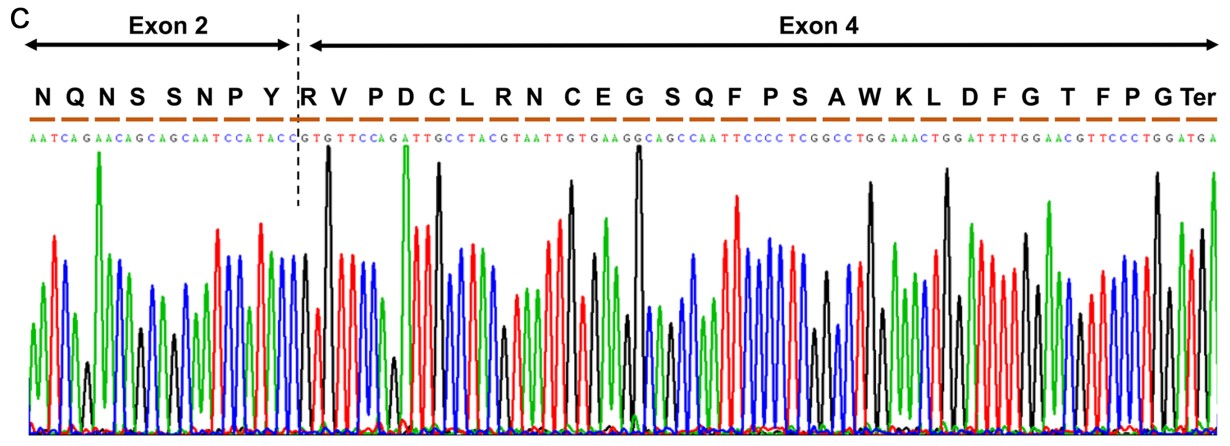

**Figure 1.** Generation of a mouse model with deletion of exon 3 (ΔExon 3) in the laminin alpha 2-chain gene (*Lama2*). (**A**) Strategy of generation of ΔExon 3 mice (dy^H^/dy^H mice) by clustered regularly interspaced short palindromic repeats (CRISPR)-Cas9. The sequences of the *Lama2* gene marked with blue indicate sgRNAs targeting sequences, ones marked with brown indicate the location of polymerase chain reaction (PCR) primers (*Lama2*-F, *Lama2*-R1, and *Lama2*-R2), respectively. (**B**) PCR analysis for genotype identification of the F2 mice. (**C**) DNA sequencing of reverse transcription (RT)-PCR products from dy^H^/dy^H muscle validated the deletion of exon 3, which resulted in a frameshift downstream sequence of *Lama2* gene.

The online version of this article includes the following source data for figure 1:

**Source data 1.** Original files for gels of polymerase chain reaction (PCR) analysis for genotype identification of the F2 mice by 2% agarose electrophoresis analysis displayed in *Figure 1B*.

**Source data 2.** PDF file containing original gels of polymerase chain reaction (PCR) analysis for genotype identification for *Figure 1B*.

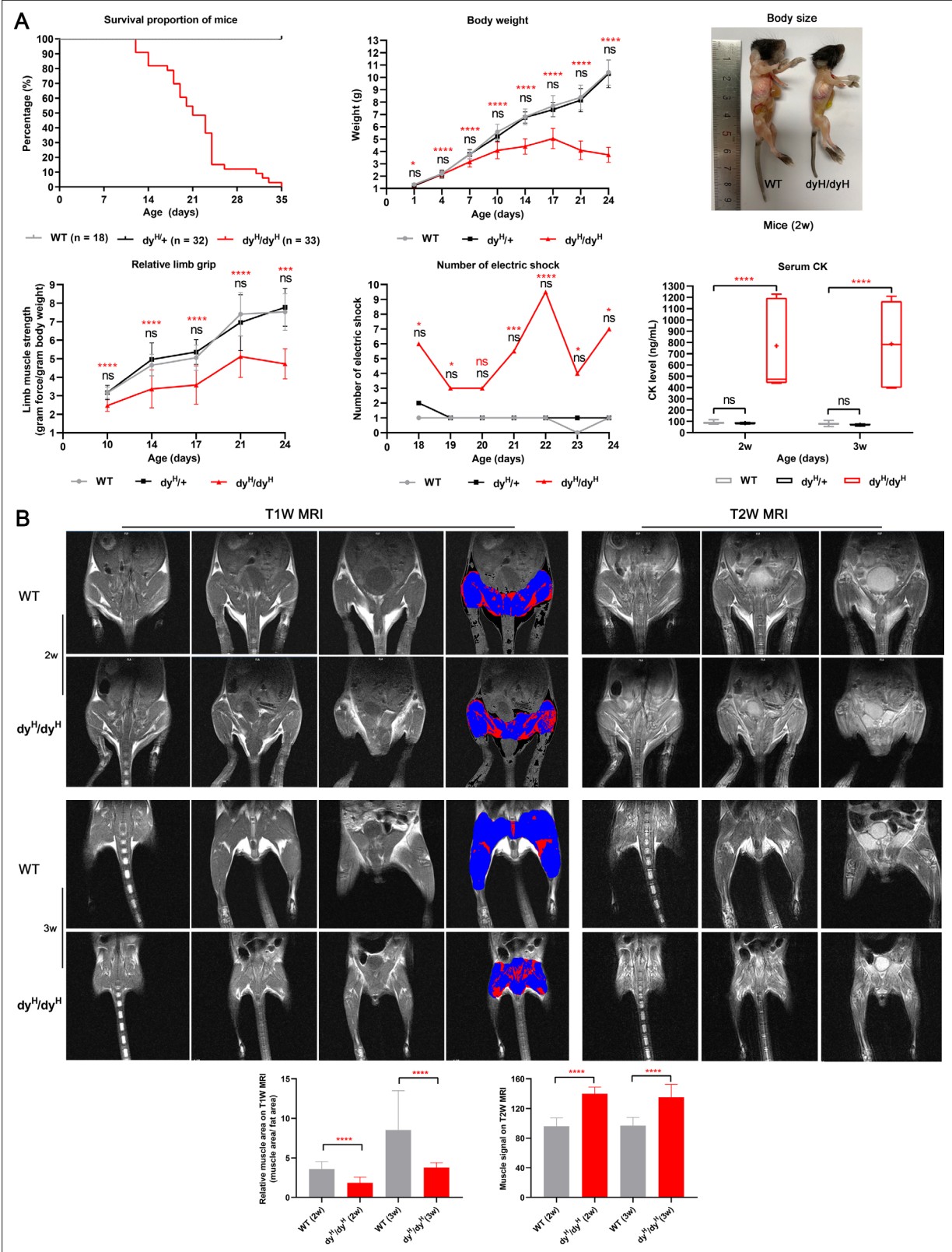

**Figure 2.** General phenotype and muscle magnetic resonance imaging (MRI) of dy^H/dy^H mice. (**A**) General phenotype of dy^H/dy^H mice. There was no significant difference in survival, body weight, muscle function (four-limb grip and the number of electric shocks on the treadmill), and serum creatine kinase (CK) levels between the wild-type (WT) and heterozygous mice at each point. Kaplan–Meier survival analysis revealed that the median survival of dy^H/dy^H mice (n=33) was 21 days (range 12–35 days), and none of the wild-type (n=18) and heterozygous (n=32) mice died during the observation

*Figure 2 continued on next page*

Figure 2 continued

period. Significant difference in body weight (g) between the WT and dy$^H$/dy$^H$ mice was marked at P7 (3.74 vs 3.16, t-test, p<0.001) (WT: n=19; dy$^H$/dy$^H$: n=20), P10 (5.58 vs 4.09, t-test, p<0.001) (WT: n=11; dy$^H$/dy$^H$: n=16), P14 (6.81 vs 4.41, t-test, p<0.001) (WT: n=12; dy$^H$/dy$^H$: n=20), P17 (7.66 vs 5.04, t-test, p<0.001) (WT: n=12; dy$^H$/dy$^H$: n=14), P21 (8.34 vs 4.09, t-test, p<0.001) (WT: n=16; dy$^H$/dy$^H$: n=20), and P24 (10.41 vs 3.73, t-test, p<0.001) (WT: n=14; dy$^H$/dy$^H$: n=13). Two-week-old dy$^H$/dy$^H$ mice could be easily identified due to their smaller size. Significant differences in the mean relative four-limb grip (force per gram body weight) between the WT and dy$^H$/dy$^H$ mice were marked at P10 (3.16 vs 2.46, t-test, p<0.001) (WT: n=11; dy$^H$/dy$^H$: n=7), P14 (4.65 vs 3.37, t-test, p<0.001) (WT: n=18; dy$^H$/dy$^H$: n=17), P17 (5.05 vs 3.58, t-test, p=0.001) (WT: n=12; dy$^H$/dy$^H$: n=10), P21 (7.41 vs 5.12, t-test, p<0.001) (WT: n=12; dy$^H$/dy$^H$: n=9), and P24 (7.53 vs 4.73, t-test, p=0.003) (WT: n=11; dy$^H$/dy$^H$: n=3). The mean number of electric shocks on the treadmill significantly increased number of the electric shocks in dy$^H$/dy$^H$ mice at P18 (t-test, p=0.010) (WT: n=13; dy$^H$/dy$^H$: n=9), P19 (t-test, p=0.012) (WT: n=13; dy$^H$/dy$^H$: n=8), P21 (t-test, p=0.003) (WT: n=13; dy$^H$/dy$^H$: n=6), P22 (t-test, p<0.001) (WT: n=13; dy$^H$/dy$^H$: n=6), P23 (t-test, p=0.017) (WT: n=13; dy$^H$/dy$^H$: n=5), and P24 (t-test, p=0.010) (WT: n=13; dy$^H$/dy$^H$: n=4). Approximately 9–10 times higher CK levels were detected in dy$^H$/dy$^H$ mice than in WT mice at P14 (WT: n=8; dy$^H$/dy$^H$: n=7) and P21 (WT: n=8; dy$^H$/dy$^H$: n=6). *p<0.05, **p<0.01, ***p<0.005, ****p<0.001, ns (p≥0.05), blue 'ns' for wild-type vs heterozygous mice, black 'ns' for vs dy$^H$/dy$^H$ mice at two ages, and orange asterisks or 'ns' for WT vs dy$^H$/dy$^H$ mice. (B) Muscle MRI in dy$^H$/dy$^H$ mice. Pelvic and hindlimb muscle MRI showed significantly smaller muscle volumes on T1-weighted MRI and significantly increased hyperintense regions on T2-weighted muscle MRI in dy$^H$/dy$^H$ mice (n=4) than in WT mice (n=4) at P14 and P21 (****p<0.001).

The online version of this article includes the following figure supplement(s) for figure 2:

**Figure supplement 1.** Body weight and muscle function analysis of dy$^H$/dy$^H$, wild-type (WT), and Het mice.

## Disruption of blood-brain barrier in dy$^H$/dy$^H$ mouse brain

The blood-brain barrier (BBB) is composed of several cell types, including microvascular endothelial cells, pericytes, astrocytes, vascular smooth muscle cells, and the basement membrane (*Villanova et al., 1997*; *Villanova et al., 1996*; *Hagg et al., 1997*). These components, along with microglia, collectively form the structure known as the neurovascular unit. Previous studies have indicated that laminin α2 is prominently expressed in astrocytes and pericytes and primarily localized to the basal lamina of cerebral blood vessels in the adult human brain (*Mohassel et al., 2018*; *Arreguin and Colognato, 2020*).

To study the molecular mechanisms underlying the brain abnormalities in the dy$^H$/dy$^H$ mice, we conducted a comprehensive analysis of laminin α2 expression on the brains of the dy$^H$/dy$^H$ (KO) and WT mice at P14 (*Figure 5*). scRNA-seq was performed to compare the transcriptional profiles between dy$^H$/dy$^H$ and WT mice brains. A total of 10,094 cells from the dy$^H$/dy$^H$ mouse brain and 10,496 cells from the WT mouse brain were captured using the 10 X Genomics platform (*Figure 5—figure supplement 1A, B*). The cell clusters within both the dy$^H$/dy$^H$ and WT brains were annotated through aggregation of the sequencing data, and classified into three major categories: neuronal cells, glial cells, and non-neuronal cells. Fifteen distinct clusters, identified with marker genes (*Figure 5—figure supplement 1C, D*), included hippocampal neurons (Hippo-neuron), glutamatergic neurons (Glu-neuron), GABAergic neurons (GABA-neuron), neuron-glial antigen 2 neurons (NG2-neuron), neuron-C5, astrocytes (Astro), cerebellum glia cells (CGC), oligodendrocytes (Oligo), microglia (Micro), vascular and leptomeningeal fibroblasts (VLF), vascular smooth muscle cells (VSM), choroid plexus cells (CPC), ependymal cells (Epen), endothelial cells (Endo), and macrophages (Macro) (*Figure 5A and B*). The comparative analysis of the proportion of each cell cluster between the dy$^H$/dy$^H$ and WT brains revealed a notable increase in GABAergic neurons (dy$^H$/dy$^H$: 5.3% vs WT: 3.1%) and a decrease in astrocytes (dy$^H$/dy$^H$: 17.1% vs WT: 20.7%) and glutamatergic neurons (dy$^H$/dy$^H$: 10.0% vs WT: 10.5%) in the dy$^H$/dy$^H$ brain (*Figure 5C*).

To ascertain which cell types expressed laminin α2, we conducted RNA sequencing analysis to measure the *Lama2* expression within each cell cluster. The results revealed that in the WT mouse brain, *Lama2* exhibited selective and robust expression in vascular and leptomeningeal fibroblasts and vascular smooth muscle cells with a low level of expression in astrocytes (*Figure 5D*). Of note, *Lama2* was predominantly expressed in astrocyte clusters 1 and 2 (*Figure 5—figure supplement 1E*). The immunofluorescence analysis found that the laminin α2 was predominantly observed on the cortical surface, and co-localized with glial fibrillary acidic protein (GFAP, a marker of astrocyte) in the WT mice at P14 (*Figure 5E*). In contrast, laminin α2 protein was absent and the expression of GFAP was reduced in the leptomeningeal of the dy$^H$/dy$^H$ mice (*Figure 5C and E*). The results suggested that laminin α2, the main component of the gliovascular basal lamina, was crucial for the leptomeningeal and astrocytes on the cortical surface.

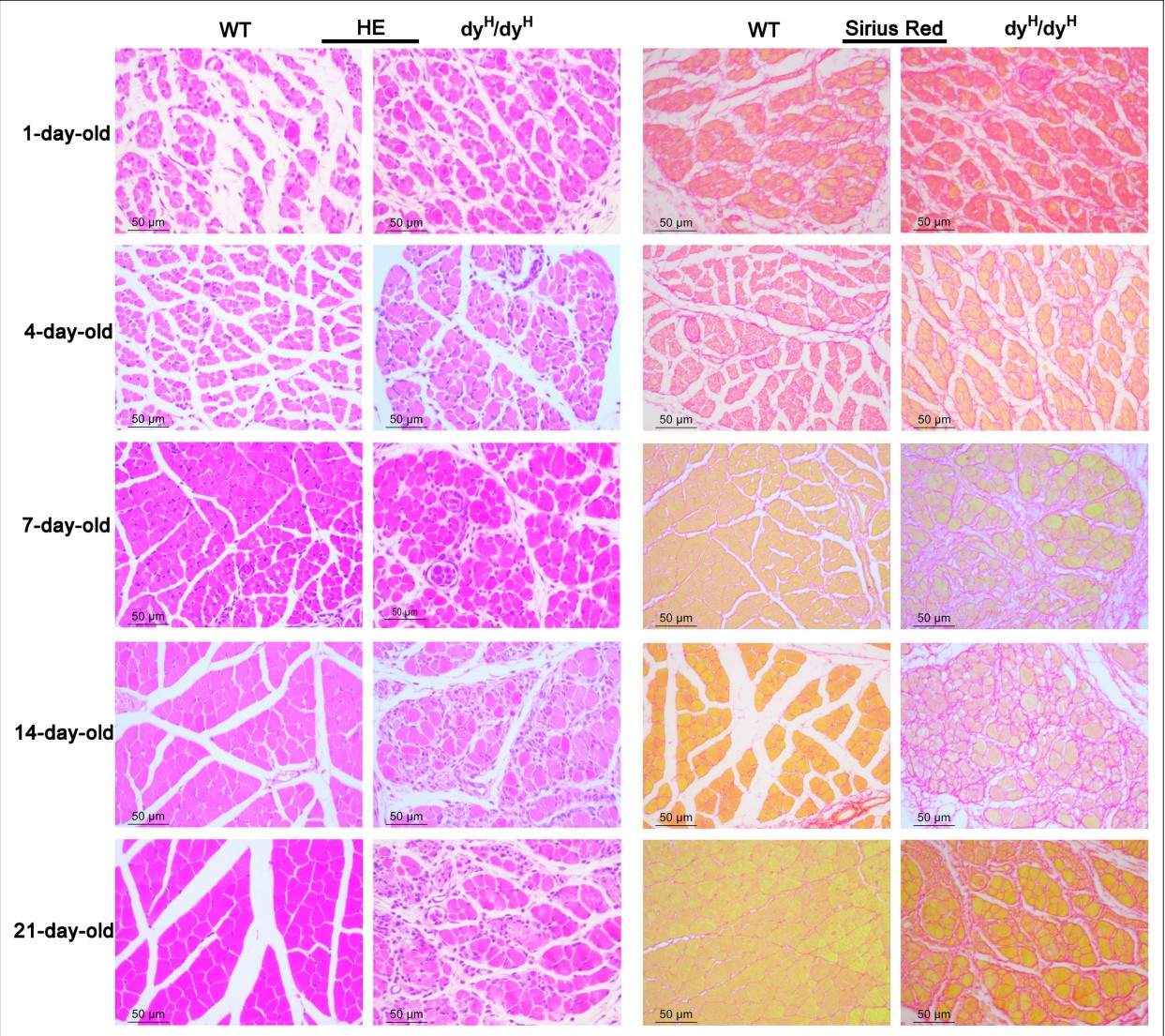

**Figure 3.** Muscle pathology with age in the biceps femoris of dy^H^/dy^H^ mice. Hematoxylin and eosin (H&E) and Sirius Red staining of the biceps femoris were compared between wild-type and dy^H^/dy^H^ mice at P1, P4, P7, P14, and P21. H&E staining showed a noticeable inequality in muscle fiber size in P7 dy^H^/dy^H^ mice, the most extreme dystrophic changes (inequality in muscle fiber size, degeneration, necrosis, and regeneration of muscle fibers), and connective tissue penetration and inflammation in P14 dy^H^/dy^H^ mice. Sirius Red staining showed increased collagen content and fibrosis in dy^H^/dy^H^ mice from P7. Scale bars: 50 μm.

The online version of this article includes the following figure supplement(s) for figure 3:

**Figure supplement 1.** Muscle pathology with age in the biceps femoris of dy^H^/dy^H^ mice.

Differentially expressed genes (DEGs) in each cell cluster between the dy^H^/dy^H^ and WT brains were analyzed to uncover the aberrant transcriptional patterns across various cell types in the dy^H^/dy^H^ mouse brain. As anticipated, *Lama2* expression was significantly diminished in vascular and leptomeningeal fibroblasts, vascular smooth muscle cells, and astrocytes in the dy^H^/dy^H^ brain, compared to the WT brain (*Figure 5D*, *Figure 5—figure supplement 1E*). Gene Ontology (GO) terms analysis revealed that the DEGs were enriched in several extracellular matrix processes within vascular and leptomeningeal fibroblasts. For vascular smooth muscle cells, the DEGs were associated with the pathway of cytoplasmic side of the membrane, axon and astrocyte projections (*Figure 5F*). The results provided potentially useful data for the disrupted structure and function of the BBB and pia mater as a consequence of *Lama2* knockout.

To delve into the alterations within the intricate network of the brain, particularly the BBB, cell-cell communications were predicted by CellChat V1.6, a tool which quantitatively inferred and

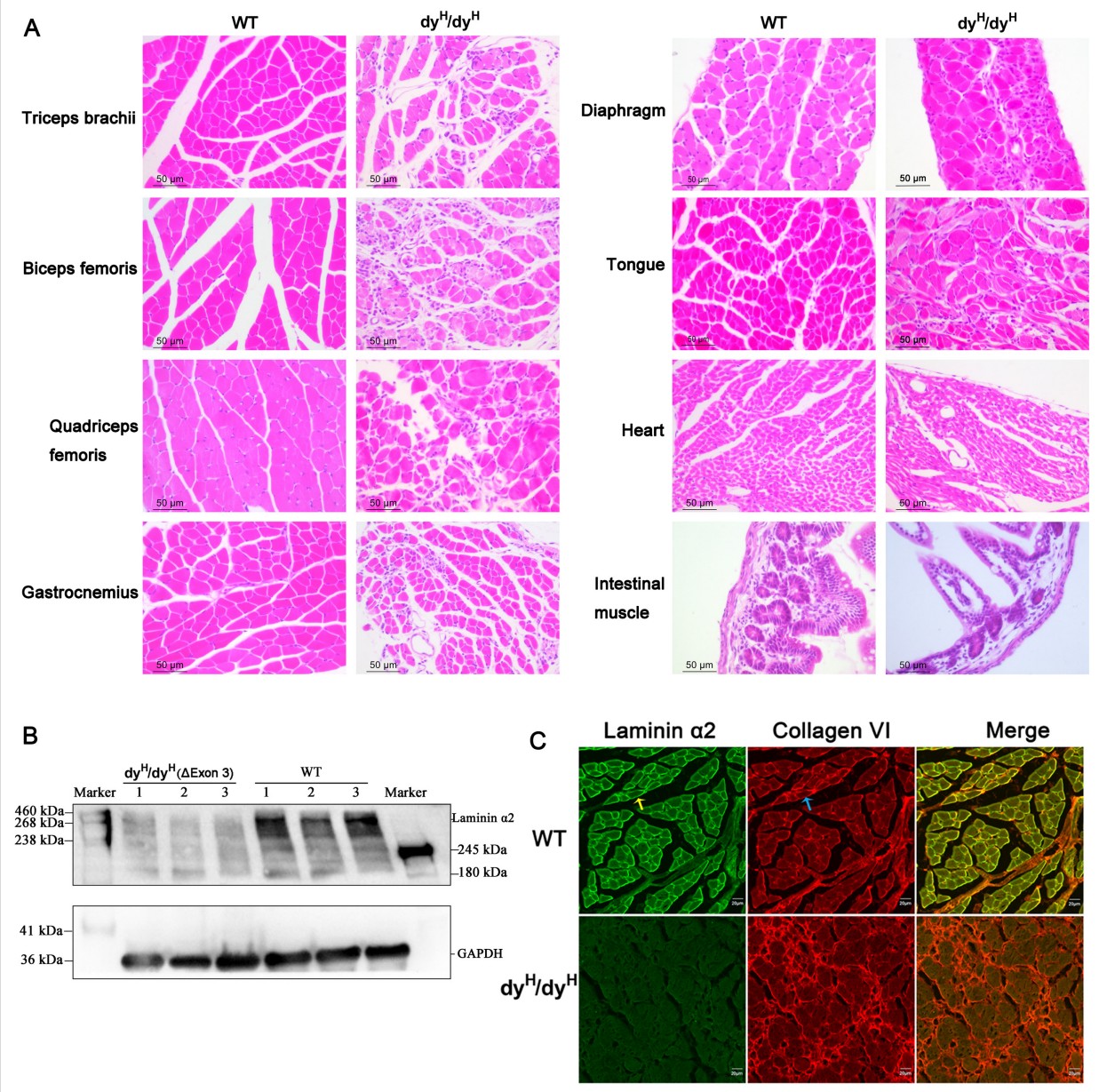

**Figure 4.** Extensive involvement with muscle pathology and laminin α2 deficiency in dy$^H$/dy$^H$ mice. (**A**) Hematoxylin and eosin (H&E) staining showed dystrophic changes in the quadriceps femoris, gastrocnemius, triceps brachii, diaphragm, and tongue muscles, but not as much as in the biceps femoris, while the heart and intestinal smooth muscles were spared. Scale bars: 50 μm. (**B**) The 300 kDa fragment of laminin α2 chain was detected in the biceps femoris of wild-type (WT) mice (n=3), but not in that of dy$^H$/dy$^H$ mice (n=3) by Western blot analysis. (**C**) Colocalization of laminin α2 chain (green fluorescence, yellow arrow) and collagen VI (red fluorescence, blue arrow) showed that they were normally located and expressed in the basement membrane in WT muscle, but laminin α2 deficiency along with increased collagen VI were observed in dy$^H$/dy$^H$ muscle.

The online version of this article includes the following source data and figure supplement(s) for figure 4:

**Source data 1.** Original files for western blot analysis displayed in *Figure 4B*.

**Source data 2.** PDF file containing original western blots for *Figure 4B*, indicating the relevant bands.

**Figure supplement 1.** Extensive involvement with muscle pathology in dy$^H$/dy$^H$ mice at P21.

analyzed intercellular communication networks from scRNA-seq data (*Jin et al., 2021*). Upon a more detailed investigation of communications among different cell populations, it was found that the distribution of communications was largely similar between the dy$^H$/dy$^H$ and WT brains. Notably, the most robust communications were observed between vascular and leptomeningeal fibroblasts and

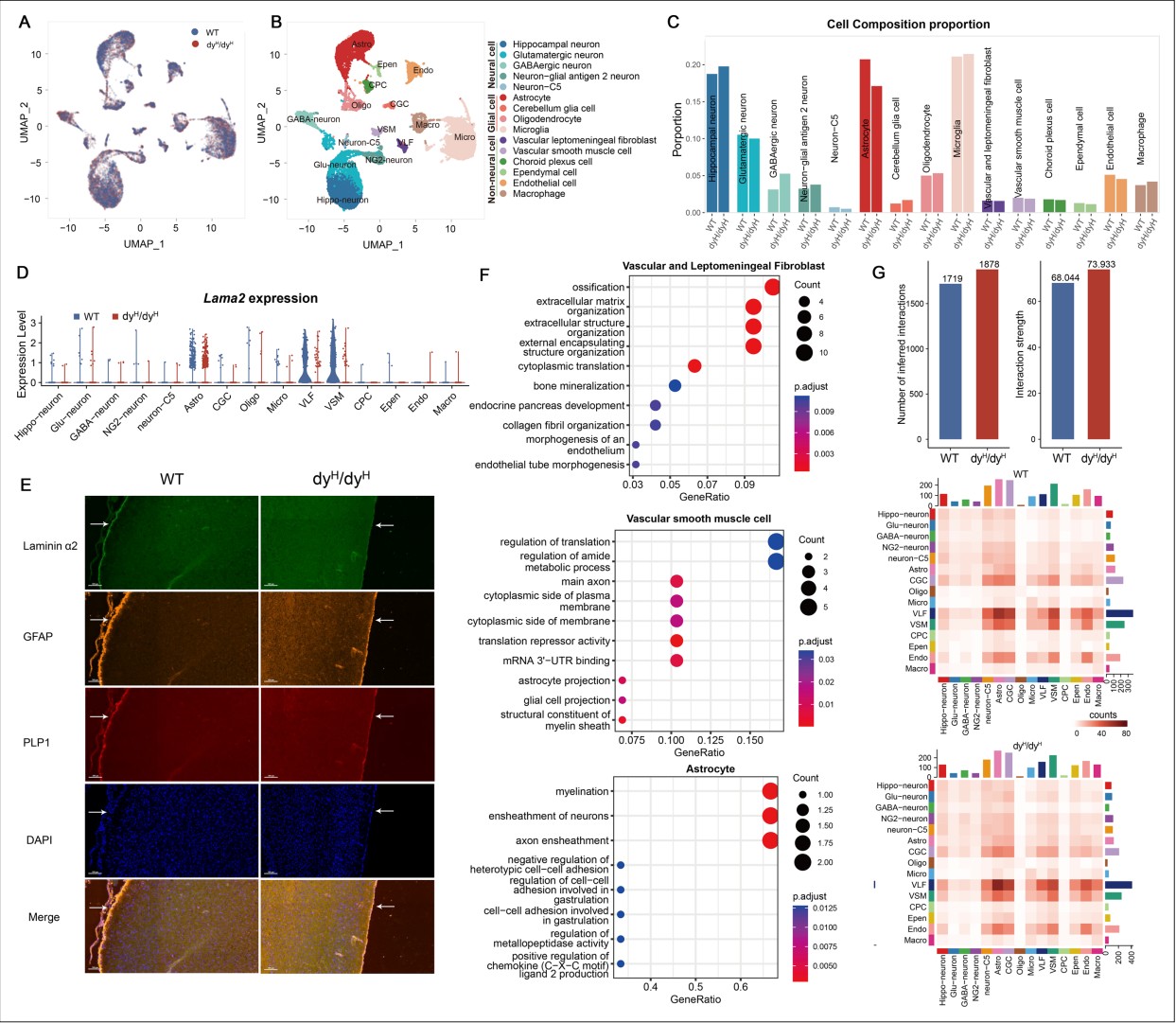

**Figure 5.** Ssingle-cell RNA sequencing (scRNA-seq) analysis of brains from dy^H/dy^H and wild-type (WT) mice. (**A**) UMAP visualization of all the cells colored by sample. (**B**) UMAP visualization of all the cells from dy^H/dy^H and WT brains colored by cluster identity. (**C**) Cell composition proportion of every annotated cell cluster in dy^H/dy^H and WT separately. (**D**) Violin plots of the expression of *Lama2* gene in WT and dy^H/dy^H in each annotated cell cluster. (**E**) Immunofluorescence of laminin α2, GFAP (astrocytes), and PLP1 on P14 mouse brain slices. DAPI was for nuclei staining. White arrows indicated the cortical surface. (**F**) Gene Ontology (GO) enrichment of differentially expressed genes in vascular and leptomeningeal fibroblasts, vascular smooth muscle cells, and astrocytes. (**G**) Cell-cell communications among cell clusters. Total number of inferred cell-cell interaction counts (left) and strength (right) in dy^H/dy^H and WT, respectively. Heatmap of ligand-receptor interaction counts in all pairwise cell clusters in WT and dy^H/dy^H brains.

The online version of this article includes the following figure supplement(s) for figure 5:

**Figure supplement 1.** Single-cell RNA sequencing (scRNA-seq) analysis of dy^H/dy^H and wild-type (WT) mouse brains.

astrocytes, and the levels of communications involving vascular and leptomeningeal fibroblasts were high (*Figure 5G*). Although the overall distribution and strength of communications were not significantly different between the dy^H/dy^H and WT brains, noticeable differences were observed in the communications associated with the laminins' pathway. Notably, the pairs of communication pathways involving *Lama2* and its receptors (integrins and Dag1) were presented only in the WT brain, whereas the ligand-receptor pairs related to *Lama1* were exclusively observed in the dy^H/dy^H brain. Moreover, several pairs demonstrated higher communication levels in the dy^H/dy^H brain than in the WT brain (*Figure 6A*). These findings suggested that the loss of communications associated with laminin α2 led to the activation of compensatory communications within other laminin pathways, providing potentially important information for the *Lama1* gene replacement therapy. Moreover, significant differences in cell-cell communications were also particularly evident in pairs involving vascular and leptomeningeal

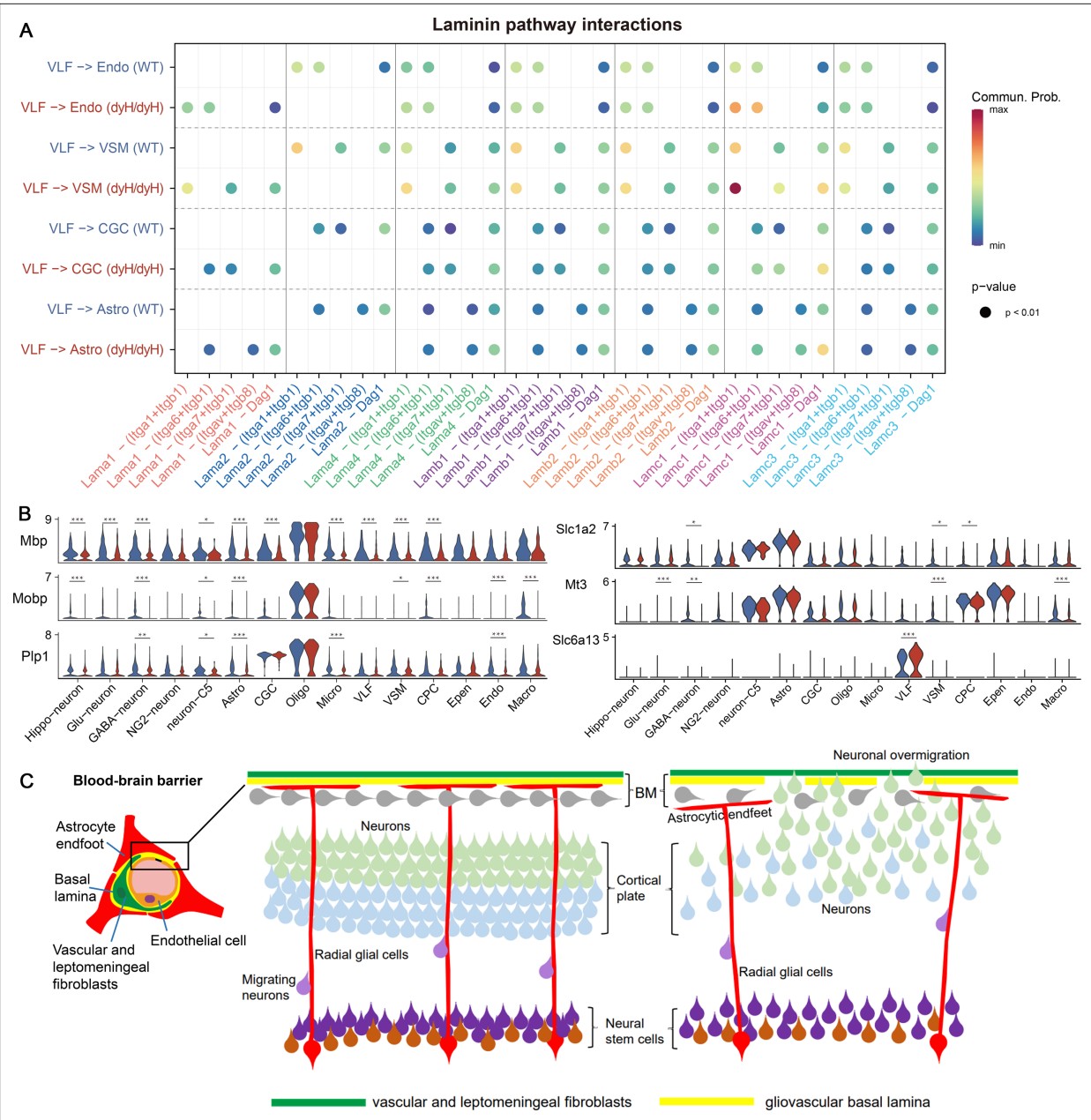

**Figure 6.** Laminin pathway interactions and hypothesis for cortical dysplasia. (**A**) The ligand-receptor pairs in laminin pathway. (**B**) Violin plots showed the expression of *Mbp*, *Mobp*, *Plp1*, *Slc1a2*, *Mt3*, and *Slc6a13* in every annotated cell cluster in dy^H/dy^H and wild-type (WT) separately. *** for p-value <0.001, ** for p-value <0.01, * for p-value <0.05. (**C**) Hypothesis that cortical dysplasia (cobblestone malformation) and neuronal over-migration in the cortex due to a defective gliovascular basal lamina of the blood-brain barrier caused by the lack of laminin α2 in *LAMA2*-MD.

fibroblast-astrocyte, vascular and leptomeningeal fibroblast-vascular smooth muscle cell, vascular and leptomeningeal fibroblast-endothelial cell, and vascular and leptomeningeal fibroblast-cerebellum glia cell pairs (**Figure 6A**).

We also found that several genes expressed differently across more than 5 cell clusters (**Figure 6B**, **Supplementary file 1**). Notably, the expressions of myelin-related genes *Mbp*, *Mobp*, and *Plp1* (**Kristiansen et al., 2009**) were significantly reduced in multiple cell clusters in the dy^H/dy^H brain compared to the WT brain, suggesting potential disruptions in the formation or development of the myelin sheath. The altered expression of the myelin-related genes supported important data for the possible mechanisms of the abnormal brain white matter observed in human *LAMA2*-MD (**Tan et al., 2021**). Also, there were some genes associated with the transmembrane ionic transportation, regulation of

synaptic transmission, and maintenance of BBB homeostasis, showing differential expression across several cell clusters (*Figure 6B*):

1. The expression of *Slc1a2*, associated with the BBB transport, was significantly decreased in GABAergic neurons, vascular smooth muscle cells, and choroid plexus cells in the dy$^H$/dy$^H$ brain compared to the WT brain.
2. The expression of *Mt3*, involved in transmembrane ionic transportation was significantly decreased in glutamatergic neurons, GABAergic neurons, vascular smooth muscle cells, and macrophages in the dy$^H$/dy$^H$ brain compared to the WT brain.
3. The expression of *Slc6a13*, involved in synaptic transmission, was notably increased in vascular and leptomeningeal fibroblasts of the dy$^H$/dy$^H$ mice.

In conclusion, these scRNA-seq analysis findings indicated that the loss of laminin α2 in vascular and leptomeningeal fibroblasts, vascular smooth muscle cells, and astrocytes might contribute to the disruption of the BBB integrity. These findings also provided potentially preliminary evidence for the dysfunction of glutamatergic and GABAergic neuronal systems and the disruptions of ionic homeostasis in the dy$^H$/dy$^H$ mouse brain.

## Impaired muscle cytoskeleton and activated muscle regeneration in dy$^H$/dy$^H$ mice

In previous studies of *LAMA2*-MD (*Nguyen et al., 2019*; *Gawlik and Durbeej, 2020*), the focus on the pathogenesis of muscular dystrophy was primarily concentrated on the secondary histopathological changes such as fibrosis, inflammation, apoptosis, and metabolism. However, the changes of muscle cytoskeleton and muscle development received relatively less attention. The muscle cytoskeletons form a complex and interconnected network, which is crucial for maintaining both cellular contractility and mechanical stability of the muscle fibers (*Henderson et al., 2017*). To underly the muscle pathology along with the damages of muscle cytoskeleton and development, as well as the secondary changes, we conducted a bulk-cell RNA sequencing (bcRNA-seq) on biceps femoris muscles obtained from the WT, heterozygote (Het, dy$^H$/+), and dy$^H$/dy$^H$ (KO) mice at P14. Principal component analysis (PCA), a technique used to reduce complex data to its principal components for visualization, demonstrated distinct patterns between the dy$^H$/dy$^H$ and WT muscle tissues (*Figure 7A*). A total of 2020 DEGs were identified, with 1136 genes upregulated and 884 genes downregulated in the KO muscles compared to the WT muscles (*Supplementary file 2*). To pinpoint the biological pathways linked to the pathogenic mechanism in KO muscle tissues, we conducted gene set enrichment analyses (GSEA) using both Gene Ontology (GO) and the Kyoto Encyclopedia of Genes and Genomes (KEGG) databases (*Huang et al., 2021*). The GO function analysis demonstrated that the DEGs were enriched in the muscle cytoskeleton and development, extracellular matrix, cell membrane, inflammation, apoptosis, and mitochondrial energy metabolism (*Figure 7B*). To shed light on the relatively less understood aspects of dystrophic pathology, we particularly focused on the DEGs related to muscle cytoskeleton and development. The result revealed a significant downregulation of several muscle cytoskeleton-related genes in the KO mice, including *Myh6*, *Myh7*, *Myl3*, *Myl2*, *Tuba8*, *Myoz3*, *Actc1*, *Mstn*, *Tppp*, *Mylk4*, *Mybpc2*, *Mrln*, *Mybph*, *Ckmt2*, *Myct1*, and *Abra* (*Figure 7B*). The abnormal expressions of muscle cytoskeleton-related proteins, including MYHC (myosin heavy chain), MYH2, desmin, and β-tubulin, were further assessed by immunofluorescence and Western blot. The immunofluorescence analysis showed a focal and increased expression of MYHC in the dy$^H$/dy$^H$ muscles (*Figure 8*, *Figure 8—figure supplement 1A*), and the significantly elevated levels of MYHC and MYH2 proteins in the dy$^H$/dy$^H$ muscles were detected by Western blot (*Figure 8*, *Figure 8—figure supplement 1B*). Desmin and β-tubulin also exhibited focal increases in the dy$^H$/dy$^H$ muscles through immunofluorescence analysis, though showed no significant differences by Western blot analysis. These results provided evidence for the impairment of the muscle cytoskeleton in the KO mice.

Further analysis of DEGs also revealed an upregulation of several myogenic regulatory factors (MRFs) genes, including *Myog*, *Myof*, *Myo5a*, *Myh4*, *Myh3*, and *Myh8* in the dy$^H$/dy$^H$ muscles. Then, we analyzed three known typical MRFs (MYOG, MYOD1, and MYF5) by western blot and immunofluorescence. The western blot analysis showed no significant differences in the total levels of MYOG and MYOD1 proteins in the dy$^H$/dy$^H$ muscles (*Figure 8*, *Figure 8—figure supplement 1B*). However, the immunofluorescence analysis demonstrated focally increased expressions of MYOG, MYOD1, and MYF5 proteins in the regions of obvious muscle regeneration in the dy$^H$/dy$^H$ muscles (*Figure 8*,

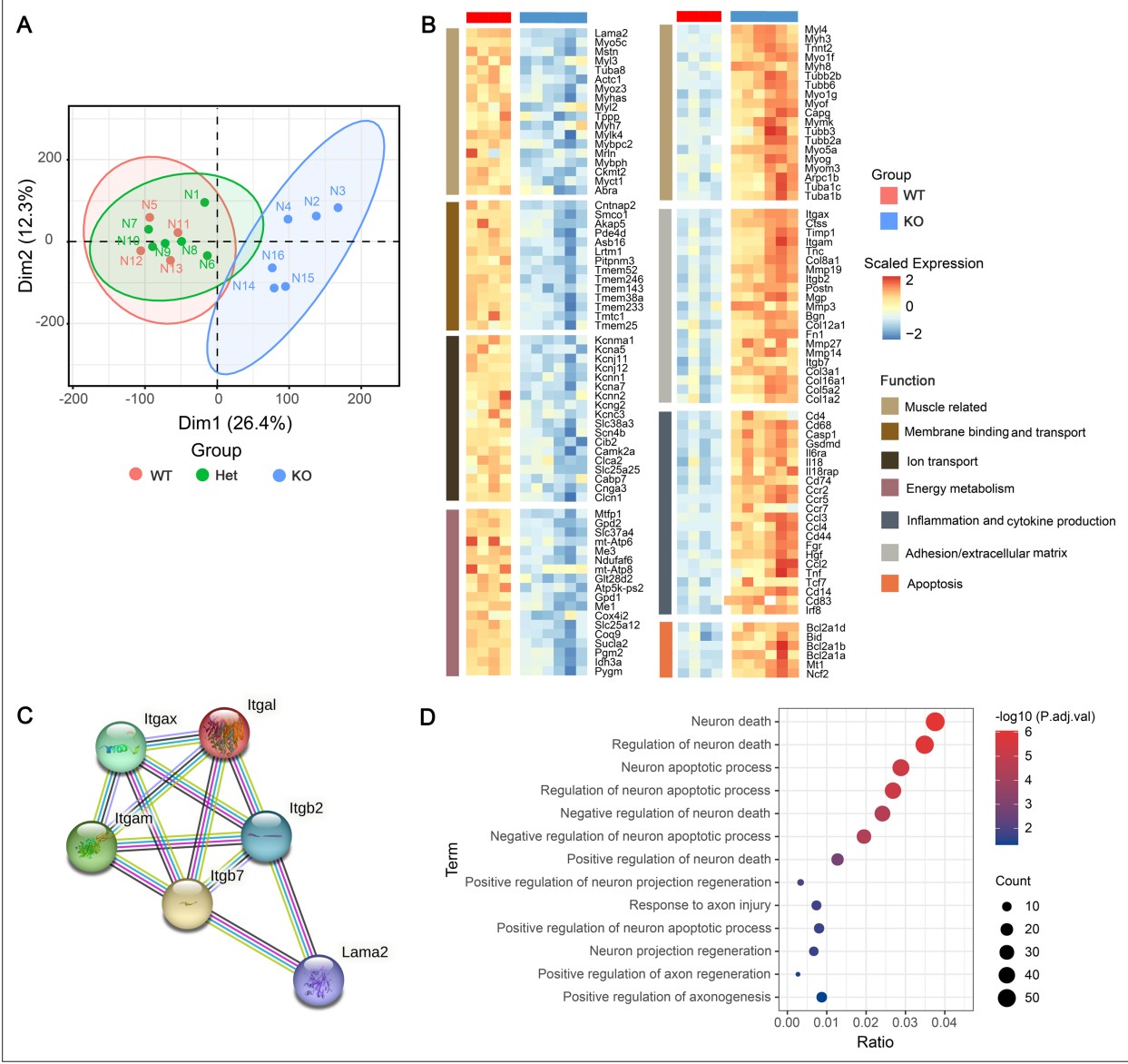

**Figure 7.** The transcriptional landscape of dy^H/dy^H (knockout, KO) mouse muscles in response to wild-type (WT) mouse muscles. (**A**) Principal component analysis for dimension reduction of differentially expressed genes (DEGs) in the biceps femoris of dy^H/dy^H and dy^H/+mice in response to WT mice. (**B**) DEGs in dy^H/dy^H muscles relative to WT muscles were enriched in adhesion/extracellular matrix, cytoskeletal proteins and muscle development, ion channel and cell membrane transport, inflammation, mitochondrial metabolism, and apoptosis. (**C**) Protein-protein interactions between *Lama2* and integrins-related genes, including *Itgal, Itgax, Itgam, Itgb2*, and *Itgb7*. (**D**) The neuron death-related signaling pathways were enriched in dy^H/dy^H muscles relative to WT muscles by Gene Ontology (GO) pathway analyses.

The online version of this article includes the following figure supplement(s) for figure 7:

**Figure supplement 1.** Immunofluorescence staining for α-dystroglycan in the biceps femoris of dy^H/dy^H mice.

---

*Figure 8—figure supplement 1A*). The results indicated the disordered muscle regeneration in the KO mice.

Considering the two major groups of receptors for LM-211, α-dystroglycan and integrins, our investigation focused on the hypothesis that genes related to *Dag1* and integrins might display differential expression (*Hohenester, 2019*; *Durbeej, 2010*; *Aumailley, 2021*). Indeed, the integrin-related genes, including *Itgal, Itgax, Itgam, Itgb2*, and *Itgb7* were upregulated in the dy^H/dy^H muscles. Furthermore, protein-protein interaction analysis demonstrated a direct interaction between Lama2 and Itgb2, as well as Lama2 and Itgb7 (*Figure 7C*). However, in contrast to integrin-related genes, no differential expression of the *Dag1* gene was observed, and the expression of α-dystroglycan protein showed no

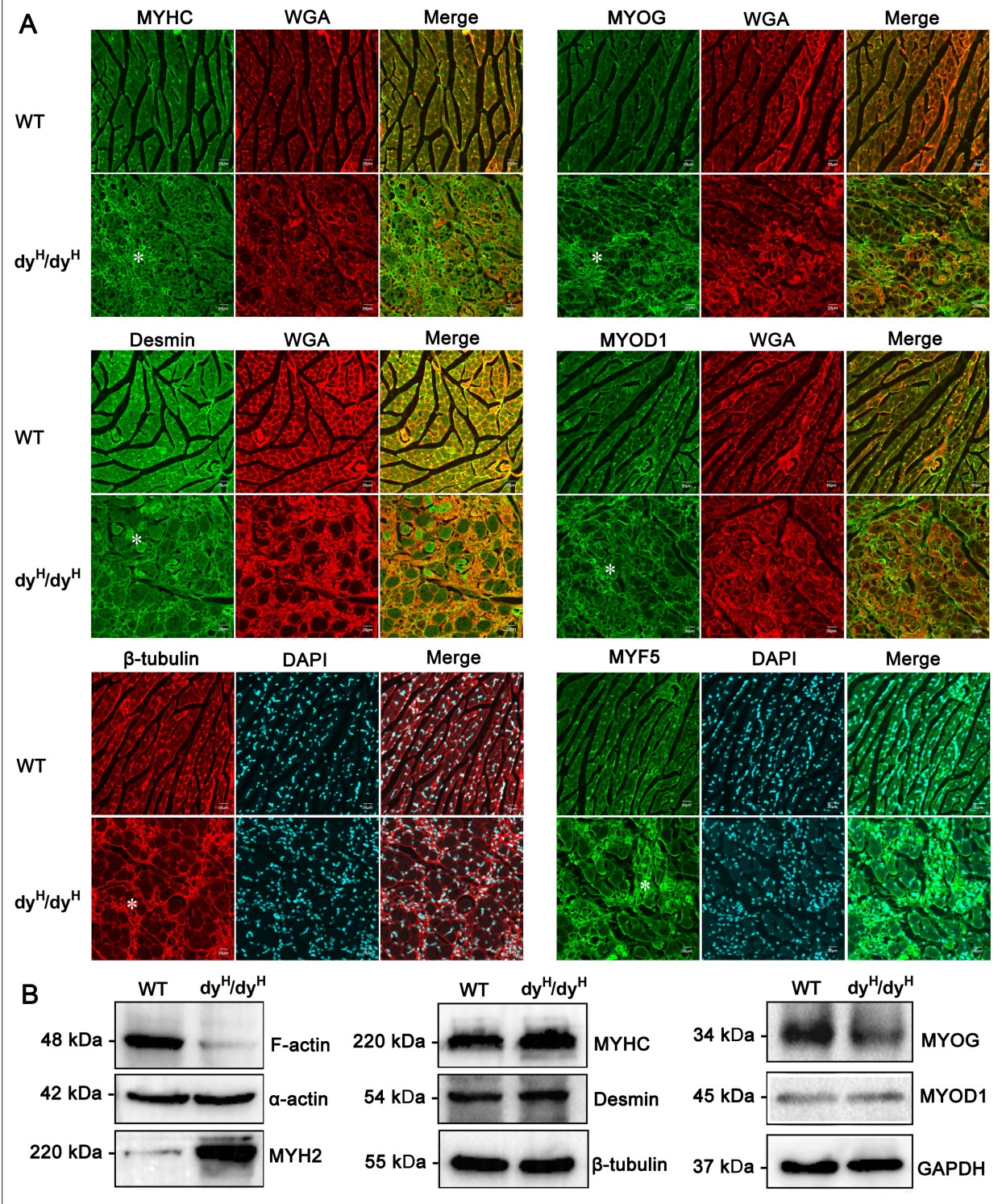

**Figure 8.** Changes of muscle cytoskeleton and development proteins in dy^H/dy^H mice. (**A**) Immunofluorescence staining for myosin heavy chain (MYHC), desmin, β-tubulin, MYOG, MYOD1, and MYF-5 in WT and dy^H/dy^H muscles at P14. MYHC, MYOG, MYOD1, and MYF-5 were focally increased, while desmin and β-tubulin showed mild decrease in the cytoplasm of hypertrophic muscle fibers as well as focal increase in developmental muscle fibers in dy^H/dy^H mice. Wheat-germ agglutinin (WGA; red) was used to visualize the muscle fibers and connective tissue. White asterisk indicated focal regions with changes. (**B**) F-actin, α-actin, MYH2, MYHC, desmin, β-tubulin, MYOG, and MYOD1 were detected by Western blot in P14 WT (n=6) and dy^H/dy^H (n=6) muscles. The levels of F-actin, MYOG and MYOD1 were significantly decreased in dy^H/dy^H muscles (p<0.05), MYH2 and MYHC were significantly increased in dy^H/dy^H muscles (p<0.05), while α-actin, desmin, and β-tubulin showed no significant difference between WT (n=6) and dy^H/dy^H (n=6) mice.

*Figure 8 continued on next page*

*Figure 8 continued*

The online version of this article includes the following source data and figure supplement(s) for figure 8:

**Source data 1.** Original files for western blot analysis displayed in *Figure 8B*.

**Source data 2.** PDF file containing original western blots for *Figure 8B*, indicating the relevant bands.

**Figure supplement 1.** Quantitative analysis of muscle cytoskeleton and development proteins in dy^H^/dy^H^ mice.

**Figure supplement 2.** Proposed pathogenic mechanisms hypothesis in the dy^H^/dy^H^ mouse model of laminin alpha 2-chain gene (*LAMA2*)-congenital muscular dystrophy (CMD).

significant difference detected by immunofluorescence (*Figure 7—figure supplement 1*). These data suggested that the muscle damage in dy^H^/dy^H^ mice might be potentially mediated by the abnormal integrin signaling.

These observations supported important data for the impaired muscle cytoskeleton and abnormal muscle development which were associated with the muscle pathology consequence of severe dystrophic changes in the dy^H^/dy^H^ mice.

## Discussion

Since the initial identification of the merosin-deficient families with *LAMA2* homozygous mutations in 1995 (*Helbling-Leclerc et al., 1995*), *LAMA2*-CMD has been the most common CMD subtype worldwide (*Sframeli et al., 2017*; *Abdel Aleem et al., 2020*; *Ge et al., 2019*). However, the underlying pathogenesis is still not completely understood and there is no effective treatment in clinic. Mouse models are essential tools for studying the pathogenic mechanism and developing therapeutic strategies for human muscular dystrophies and congenital myopathies (*Sztretye et al., 2020*; *van Putten et al., 2020*). Here, we generated a *LAMA2*-CMD mouse model by utilizing a CRISPR/Cas9 technology and based on the frequent disease-causing genetic variants in human patients. We described the phenotype of the new *LAMA2*-CMD mouse model, and studied transcriptomic changes to better understand the molecular mechanism.

Recently, cortical malformations, including polymicrogyria, lissencephaly-pachygyria, and cobblestone, most severe in the occipital lobe, such as occipital pachygyria, have been thought to be an important manifestation of the brain pathology in *LAMA2*-MD (*Jayakody et al., 2020*; *Camelo et al., 2023*). During brain development, laminins are enriched at ventricular and pial surfaces and the BBB to mediate radial glial cells (RGCs) attachment for neuronal migration. The abnormal neuronal migration in specific areas of the cortex due to the lack of RGCs attachment to the gliovascular basal lamina and consequent gaps in that membrane, which was associated with the cortical malformations (*Severino et al., 2020*). In previous studies, it was reported that laminin α2 deficiency could result in an impaired gliovascular basal lamina of BBB, including altered integrity and composition of the endothelial basal lamina, abnormal pericytes and astrocytic endfeet (*Devisme et al., 2012*; *Menezes et al., 2014*). In our studies, the selectively and highly expressed *Lama2* gene in vascular and leptomeningeal fibroblasts, and vascular smooth muscle cells, and the predominantly expressed laminin α2 protein on the cortical surface in the WT brain are correspondent to its role in the BBB. The corresponding deficiency of both *Lama2* gene and laminin α2 protein in the dy^H^/dy^H^ brains could provide the precondition basis for the impaired gliovascular basal lamina of BBB. Laminins anchor to integrins on the cell surface to form laminins-integrins pathways. As reported, the aberrant BBB function was also associated with the adhesion defect of alpha7 integrin subunit in astrocytes to laminins in the *Itga7*^-/-^ mice (*Chen et al., 2023*). In this study, loss of communications involving the laminins' pathway between laminin α2 and integrins were predicted between vascular and leptomeningeal fibroblasts and astrocytes in the dy^H^/dy^H^ brain, providing more evidence for the impaired BBB due to laminin α2 deficiency. Though the cortical malformations were not found in the dy^H^/dy^H^ brains by MRI analysis, probably due to the small volume within 1 month old, the changes in transcriptomes and protein levels provided potentially useful data for the hypothesis of the impaired gliovascular basal lamina of the BBB, which might be associated with occipital pachygyria in *LAMA2*-CMD patients (*Figure 6C*).

Seizures have been thought as a main symptom of brain dysfunctions in *LAMA2*-MD, which might be associated with cortical malformation (*Sarkozy et al., 2020*; *Huang et al., 2023*). However, the pathogenesis of seizures in *LAMA2*-MD remains unclearly known. The occurrence of seizures was

thought to be related to the aberrant neuronal network formation and the imbalance of excitability and inhibitory neuronal network and ionic homeostasis (*Barkovich et al., 2015*; *Jayakody et al., 2020*). This study provided some potential clues for the pathogenesis with the DEGs such as *Slc1a2*, *Mt3*, and *Slc6a13*, which were associated with the glutaminergic- and GABAergic neuronal systems and the ionic homeostasis data. However, the mechanisms still need further study.

Compared with other *Lama2*-deficient mice, including dy/dy, dy$^{2J}$/dy$^{2J}$, dy$^{3k}$/dy$^{3k}$, and dy$^W$/dy$^W$, the phenotype of the dy$^H$/dy$^H$ mice was similar to that of the dy$^{3k}$/dy$^{3k}$ mice, presenting with a very severe muscular dystrophy (*Supplementary file 3*). To assess the development of muscular dystrophy from early to late postnatal periods to better understand the pathogenic processes in human *LAMA2*-CMD, we observed the muscle pathology at different ages and stages. In this study, we observed significant fiber size variations in the muscles of dy$^H$/dy$^H$ mice as early as P7. The typical muscle pathological features with marked variation in fiber size, muscle fiber degeneration, necrosis, and regeneration accompanying infiltration of inflammatory cells and fibrosis accelerated until P14 and then decelerated until they expired. Muscle inflammation and fibrosis assessed by muscle MRI was previously reported in the dy$^w$/dy$^w$ mice at the age of 7 weeks (*Vohra et al., 2015*). In this study, muscle MRI performed earlier (age of 2–3 weeks) provided the corresponding changes of muscle pathology at the early stage. We observed a decreased muscle volume due to muscle wasting on T1-weighted MRI and abnormal hyperintense pixels for inflammation on T2-weighted MRI in the dy$^H$/dy$^H$ mice. The results were consistent with muscle pathology and indicated that muscle MRI could be a useful method for monitoring changes in muscle pathology of muscular dystrophies in preclinical research (*Porcari et al., 2020*; *Decostre et al., 2013*).

Muscle pathological events of limb muscles, including triceps brachii, biceps femoris muscle, quadriceps femoris, and gastrocnemius from the early postnatal stage to the last stage could provide a better understanding of the changes of muscle damage and pathology with disease progression (*Mehuron et al., 2014*; *Durbeej, 2015*; *Gawlik et al., 2017*). In addition to the limb muscles, the diaphragm and tongue muscles were involved, indicating extensive muscle dystrophic change in the dy$^H$/dy$^H$ mice, which was consistent with those previously observed in the dy$^{3k}$/dy$^{3k}$ mice (*Gawlik et al., 2019*). The extensive involvement of muscle pathology was associated with low body weight, muscle wasting, and severe generalized weakness which resulted in respiratory failure, feeding difficulty, and the ultimate fatality. These pathological findings and muscle weakness progression in the dy$^H$/dy$^H$ mice were closely resembling those seen in human *LAMA2*-CMD (*Zambon et al., 2020*; Tan D et al., 2021; *Sarkozy et al., 2020*; *Jain et al., 2019*).

Muscle cytoskeleton network is essential for the processes of muscle contraction, force transmission, adaptation of cell shape, cell division and adhesion (*Henderson et al., 2017*). For example, F-actin and microtubules regulate cellular elasticity and mechanical contraction (*Stricker et al., 2010*; *Hohmann and Dehghani, 2019*), MYHCs provide chemical energy by hydrolysis of ATP (*Lewis and Ochala, 2023*), and desmin is associated with the mechanical integrity and mitochondria positioning (*Agnetti et al., 2022*). Interestingly, we identified a group of differentially expressed genes (DEGs) related to muscle cytoskeleton and development at P14, when the most extensive muscle dystrophic changes were observed in our dy$^H$/dy$^H$ mice. It was also reported that muscle cytoskeleton proteins such as myosin-4, titin, actin, myosin-1, myosin-3, and myosin-8 were dysregulated in the dy$^{3k}$/dy$^{3k}$ mice (*de Oliveira et al., 2014*). The dysfunction of cytoskeleton network-related proteins, secondary to laminin α2 deficiency, was speculated to probably impair the muscle structure and contractile function in dy$^H$/dy$^H$ mice, and be associated with the degeneration and necrosis of muscle fibers (*Figure 8—figure supplement 2*).

In addition to the degeneration and necrosis of muscle fibers in the dy$^H$/dy$^H$ mice, we also observed active fiber regeneration, which was associated with the abnormal expression of the muscle development-related genes and proteins. In previous reports, muscle development-related genes, such as *MyoD*, *Myh3*, and *Myof5* were upregulated in *LAMA2*-CMD mouse models (*Mehuron et al., 2014*; *Onofre-Oliveira et al., 2012*). Consistent with the previous reports, the proliferation and differentiation-related genes such as *Myog*, *Myof*, *Myh3*, and *Myh8* were upregulated in the biceps femoris of the dy$^H$/dy$^H$ mice at P14 in our current study. Moreover, MYOG, MYOD1, and MYF5 were increased in discrete areas of the dy$^H$/dy$^H$ muscles at P14 (*Figure 8A*), but the overall levels of MYOG and MYOD1 were decreased by Western blot (*Figure 8B*). These findings indicated the regeneration of muscle fibers and impaired regeneration occurred simultaneously (*Figure 8—figure supplement*

2). However, the mechanisms of the impaired muscle regeneration due to laminin α2 deficiency require further investigation.

Lm-211 connects with several extracellular matrix proteins to form basement membrane and links to the cell membrane, which provides cell microenvironment to maintain the stability of muscle fibers, regulates the transmission of mechanical force, and the development and regeneration of muscle fibers (*Zhang et al., 2021*). In previous reports, the expression of extracellular matrix proteins such as Col1a1, Col6α2, collagen III, fibronectin, periostin, galectin-1, and biglycan were extensively increased in *LAMA2*-CMD mouse models (*Mehuron et al., 2014*; *Gawlik et al., 2019*; *de Oliveira et al., 2014*). As expected, in the dy$^H$/dy$^H$ mice, following laminin α2 deficiency, extracellular matrix proteins such as collagen VI and laminin α1, and several extracellular matrix protein-related genes such as *Col1a2*, *Col5a2*, *Col3a1*, *Tnc*, *Fn1,* and *Ctss* were upregulated. These provided evidence for the compensatory hyperplasia of the extracellular matrix resulted in the pathologic fibrosis, as well as the effective treatment strategies of targeted regulation of specific components of extracellular matrix (*Ahmad, 2021*).

## Limitations

This study has reported on a new mouse model for *LAMA2*-CMD generated by CRISPR-Cas9 and provided potentially useful transcriptomics data regarding the abnormal muscle and brain with some limitations. Even though RNA-seq and scRNA-seq have been performed, the data of scRNA-seq are still insufficient due to the limited number of mouse brains. This study has provided potentially important information for the molecular pathogenetic mechanisms of muscular dystrophy and brain dysfunction for *LAMA2*-CMD, however, some related functional experiments have not been further performed. Moreover, due to making sections with PFA before muscles isolated, and not from fresh-frozen tissue, there have been big gaps in the sections which do affect the histology of skeletal muscle to some extent. Grip strength measurements used in this study are considered error-prone and do not give an accurate measurement of muscle strength, which would be better achieved using ex vivo or in vivo muscle contractility studies. Finally, due to the limited number of animal samples for the Power analysis, the statistical errors and conclusions might be affected.

In summary, the present study provided a novel mouse model dy$^H$/dy$^H$ for human *LAMA2*-CMD. This was achieved by targeted deletion of exons 3, a frequent mutation region found in *LAMA2*-CMD patients. The dy$^H$/dy$^H$ mouse presented the phenotype with severe muscular dystrophy. The detailed data involving motor function, muscle and brain pathologies, and transcriptomics changes in the dy$^H$/dy$^H$ mice would provide better understanding of the disease process of *LAMA2*-CMD. This mouse model serves as a valid disease model for further investigation into the underlying pathogenesis of human *LAMA2*-CMD and developing effective therapeutic strategies.

## Materials and methods
### Generation of a novel dy$^H$/dy$^H$ knockout mouse with ΔExon 3 at *Lama2* locus

All studies conducted on the mice were approved by the Animal Ethics Committee of Peking University First Hospital (J202027). All mice were housed and handled according to the guidelines of the Care and Use of Laboratory Animals (NIH Publication, 8th Edition, 2011). C57BL/6 mice were used and maintained on a congenic background.

To target the exon 3 of the *Lama2* gene (gene ID 16773), a pair of sgRNAs (5'-ggctgtgtatcactaattccagg-3', 5'-atggatcaagatcctatagaagg-3') targeting intron 2 and intron 3 of the *Lama2* gene were designed (*Figure 1A*). The pCS-4G vectors containing sgRNAs and Cas9 mRNA were coinjected into 270 C57BL/6 zygotes which were subsequently implanted into pseudopregnant mice, and 33 pups were born. Nine (3♀, 6♂) of them were genetically identified as heterozygotes with a heterozygous 1625-base pair (bp) deletion at genomic DNA level, and one was selected as F0 founder which was backcrossed with C57BL/6 mice to produce generation F1 mice. F2 mice generated from the crossed heterozygous F1 mice were genetically identified and used for the following studies. The genotypes were identified by two polymerase chain reaction (PCR) amplifications with primers (*Figure 1A*, *Supplementary file 4*) and PCR products DNA sequencing. The PCR–amplification conditions were as follows: 95°C for 3 min, followed by 30 cycles at 95°C for 15 s, annealing at 62°C for 20 s and elongation at 72°C for 2 min, and a final step at 72°C for 7 min.

The expected sizes observed by 2% agarose electrophoresis analysis were a 1999 bp fragment for the wild-type allele and a 374 bp fragment for the mutant-type allele in the first PCR– amplification, and a 571 bp fragment for the wild-type allele and no fragment for the mutant-type allele in the second PCR amplification (*Figure 1B*, *Supplementary file 4*). The F2 mice showing PCR products with a 1999 bp fragment in PCR1 along with a 571 bp fragment in PCR2 were wild-type (WT). Those having a 1999 bp fragment and a 374 bp fragment in PCR1 along with a 571 bp fragment in PCR2 were dy$^H$/+ (Het). Those having a 374 bp fragment in PCR1 along with no fragment in PCR2 were homozygote knockout (KO), dy$^H$/dy$^H$ mice (*Figure 1B*).

Then, genotype identification was further analyzed by reverse transcription (RT)–PCR. Total RNA was isolated from biceps femoris of WT and dy$^H$/dy$^H$, and was reverse transcribed to first strand cDNA using the Reverse Transcription System (A3500, Promega, Madison, Wisconsin, USA). RT–PCR with the forward primer (5'-TGCTTCGAATGCACTCATCACAAC-3') and the reverse primer (5'-GATATTGT AGAGGGTCAGGCACTCC-3') was performed to amplify *LAMA2* cDNA exons 2–4. The RT–PCR ampli-fication conditions were as follows: 95°C for 5 min, followed by 35 cycles at 95°C for 30 s, annealing at 56°C for 20 s and elongation at 72°C for 50 s, and a final step at 72°C for 5 min. The RT–PCR products were analyzed by DNA sequencing, and the deletion of exon 3, which resulted in a frameshift down-stream sequence of *Lama2* gene, was confirmed at cDNA level in dy$^H$/dy$^H$ mice (*Figure 1C*).

## Single-cell RNA sequencing and data process

Two total brains were extracted from 21 days of age dy$^H$/dy$^H$ and WT mouse. Single-cell RNA-seq libraries were constructed using the Chromium Single Cell 3' Reagent Kit (10 x Genomics) according to the manufacturer's protocol on the brain tissue. Sequencing was performed on Illumina platform. The raw data were processed by CellRanger to perform sample demultiplexing, barcode processing and single cell 3' gene counting. The cDNA reads were aligned to the mm10 pre-mRNA reference genome. Only confidently mapped reads with valid barcodes and unique molecular identifiers were used to generate the gene-barcode matrix. Further analyses for quality filtering were performed using the Seurat V4.3 R package. Cells, which have unique feature counts over 3000 or less than 200 or have >5% mitochondrial counts, were filtered. After quality filtering and removing unwanted cells from the dataset, we normalized the data by the total expression, multiplied by a scale factor of 10,000 and log-transformed the result, then we performed cell clustering, gene expression visualiza-tion, marker genes identification, and differential expression analysis. The threshold for differentially expressed genes was set as |log2FoldChange|>0.5 and p-value <0.05. Then, clusterProfiler V4.8 was used for GO and KEGG enrichment of the differentially expressed genes. CellChat V1.6 was used to analyze the cell-cell communications between cell clusters and compare the differences in cell-cell communications between dy$^H$/dy$^H$ and WT.

## RNA isolation and RNA sequencing

The biceps femoris was obtained from 14-day-old mice of WT (n=4), Het (n=6), and dy$^H$/dy$^H$ (KO) (n=6). Total RNA was isolated from the biceps femoris using TRIzol (Invitrogen, Carlsbad, CA). The RNA samples were submitted to CapitalBio (https://www.capitalbiotech.com) (2 × 150 bp reads) was performed on successful RNA libraries using the Illumina HiSeq X-Ten platform. During the exper-iment, investigators were blinded to the samples' information. The quality of raw reads was first assessed using FastQC. After filtering out low-quality bases and adaptors using FastP, reads were mapped to the mouse genome assembly GRCh38 (Mus_musculus.GRCm38.dna.toplevel.REF.fa) using Hisat2. Samples were subjected to quality control by examining the percentage of reads uniquely mapping to the genome, the percentage of reads mapping to known protein-coding sequences, and the number of genes with 90% base coverage. Gene fusions were identified by mapping reads to the mouse genome using StringTie. The additional information about the hidden correlations within obtained dataset was extracted by principal component analysis using dimension reduction which reduces the dimensionality of the original data matrix retaining the maximum amount of variability. Differentially expressed genes (DEGs) were identified by counting the number of reads mapping to each gene from Ensembl 96 using featureCounts and StringTie. Transcripts per million (TPM) were analyzed using Stringtie software. The R package DESeq2 was used to detect DEGs and normalize the read count. The Pearson correlation of each sample showed that all samples were highly correlated. The R package clusterProfiler was used for GO function (http://www.geneontology.org/) and GSEA

KEGG analysis. Additionally, Gene Set Variation Analysis (GSVA) was employed for GO function and KEGG pathway analysis using the R package GSVA, and the limma package (version 3.25.15; bioinf.wehi.edu.au/limma) was used to detect the differentially enriched functions and pathways ($|\log2FC|$ ≥ 1, p-value <0.05). A clustering dendrogram was used to display the results of dynamic tree cutting and merging.

## Muscle pathology, immunofluorescence, and immunohistochemistry

Skeletal muscles (biceps femoris, quadriceps femoris, and triceps brachii) were isolated from WT and $dy^H/dy^H$ mice at P1, P4, P7, P14, and P21. Gastrocnemius muscle, diaphragm, heart, tongue, and intestinal smooth muscles were isolated from WT and $dy^H/dy^H$ mice at P14. Muscle tissues were embedded in optimal cutting temperature compound (Tissue Tek, Torrance, CA) and frozen in liquid nitrogen. Seven µm thick transverse cryosections were stained with hematoxylin and eosin (H&E, Solarbio, Beijing, China), and Sirius Red or subjected to immunostaining. Immunofluorescence was performed according to standard procedures with antibodies against the N-terminus of the laminin α2 chain (rat monoclonal, 4H8-2, Sigma, Saint Louis, USA), the laminin α1 chain (rat monoclonal, MAB4656, R&D Systems, Minneapolis, USA), myogenic differentiation antigen 1 (MYOD1) (mouse monoclonal, ab16148, Abcam, Cambridge, UK), myogenin (MYOG) (mouse monoclonal, MAB66861, R&D Systems, Minneapolis, USA), myogenic factor 5 (MYF5) (mouse monoclonal, MAB4027, R&D Systems, Minneapolis, USA), myosin heavy chain (MYHC) (mouse monoclonal, MAB4470, R&D Systems, Minneapolis, USA), desmin (mouse monoclonal, MA5-15306, Invitrogen, CA), CD68 (rabbit IgG, BA3638, Boster, CA), and clone IIH6C4 for α-DG (mouse monoclonal, 05–593, Merck Millipore, Darmstadt, Germany). The secondary antibodies were goat anti-rat IgG 488, goat anti-mouse IgG 488/594, and goat anti-rabbit IgG 488/594 (Abcam, Cambridge, UK).

The slides for H&E, Sirius Red, and immunohistochemistry staining were observed by Leica microscopy (DFC295, Wetzlar, Germany) with LAS V4.12, and the slides for immunofluorescence were imaged by a confocal microscope with FV3000 system (Olympus FluoView FV10i, Tokyo, Japan).

## Western blot analysis

Total protein was extracted from mouse muscle tissues using RIPA solution (Plygen, Beijing, China). The protein concentration was determined by a BCA protein assay (Thermo Fisher Scientific Inc, Waltham, MA). Then, denatured proteins were separated by sodium dodecyl sulfate polyacrylamide gels (SDS–PAGE) and transferred to a polyvinylidene fluoride membrane. Electrochemiluminescence was used to observe the bands with an imaging system (molecular imager, ChemiDoc XRS, Bio–Rad, CA). The densities of the bands were determined semiquantitatively by ImageJ software (NIH, Bethesda, MD). Equal protein loading of blots was confirmed by immunoblotting of GAPDH (rabbit monoclonal, #2118, Cell Signaling Technology, Danvers, USA), except β-tubulin (mouse monoclonal, 86298, Cell Signaling Technology, Danvers, USA) used as equal protein for laminin α2. The results were represented as fold of change over the control (wild-type group) value. The antibodies used in this study were as follows: laminin α2 chain (rat monoclonal, 4H8-2, Sigma, Saint Louis, USA), MYOD1 (mouse monoclonal, ab64159, Abcam, Cambridge, UK), MYOG (mouse monoclonal, MAB66861, R&D Systems, Minneapolis, USA), myosin heavy chain 2 (MYH2) (rabbit monoclonal, ab124937, Abcam, Cambridge, UK), MYHC (mouse monoclonal, MAB4470, R&D Systems, Minneapolis, USA), desmin (mouse monoclonal, MA5-15306, Invitrogen, CA), α-actin (rabbit polyclonal, 23660-I-AP, Proteintech, Rosemont, USA), F-actin (rabbit polyclonal, bs-1571R, Bioss, Rosemont, USA). Then, the blots were incubated with horseradish peroxidase-conjugated secondary antibodies (Cell Signaling Technology, Danvers, USA).

## Four limbs grip strength test

Muscle strength measurements of the four limbs of mice were performed from P10-P24 using a grip strength meter and SuperGSM software (Shanghai XinRuan Information Technology Co., Ltd.). All four paws of each mouse were allowed to grasp the grid attached to the grip strength meter. After obtaining a good grip, the mouse was pulled away from the grid until the grasp broke. The test was repeated three times for each mouse and the mean value was calculated. The results are presented as normalized strength (gram force per gram body weight) (*Elbaz et al., 2012*). Four-limb grip strength

was measured by one person for all trials due to the outcome possibly being highly variable between experimenters.

### Treadmill exercise protocol

The mice were forced to run on a motorized treadmill for training prior to the experiment once a day from P13 to P17. The experiment was performed beginning on P18. The $dy^H/dy^H$ mice that were too fatigued to run were removed from the experiment. The exercise load consisted of 5 min break and interval running cycles for 25 min at a speed of 1 meter/min for 20 s and 2 meters/min for 20 s, with a 0° inclination. Motivation to run was induced by applying 1.0 mA electric foot shocks, one set of electric foot shocks with a 10 s duration. The gait and number of electric shocks were observed.

### Determination of the serum CK levels

Approximately 200 µL of blood from each mouse was collected, centrifuged at 4000 rpm for 10 min at 4°C, and analyzed for CK levels using a mouse CK ELISA Kit (Nanjing Herb-Source Bio-Technology CO., Ltd, czy24506). To determine the CK levels, the serum samples were diluted stepwise, 5-, 20-, and 50-fold because the CK level was beyond the linear range, and the CK level was then measured and recorded at the highest dilution.

### Muscle MR acquisition

Siemens TIM Trio 3.0T MRI scanner (Siemens, Erlangen, Germany) was used to detect changes in the muscles. The mice were anesthetized by intraperitoneal injection of 5% chloral hydrate (7 mL/kg), and then placed prostrate on a holder bed with the hip and hindlimbs moved into the center of a small animal-specific coil. High-resolution T1-weighted and T2-weighted MRI of the hip and hindlimb muscles were acquired under optimized imaging parameters (*Supplementary file 5*). Relative muscle area (muscle area/fat area) was quantified on T1-weighted MRI, and heterogeneity was quantified in muscles on T2-weighted MRI by averaging the mean intensities in 8–11 regions of interest (ROIs) in 3 slices using ImageJ software (NIH, Bethesda, MD) (*Iyer et al., 2020*). Images were converted to Digital Imaging and Communication in Medicine (DICOM) format using syngo MR B17 software (Siemens, Erlangen, Germany).

### Statistical analysis

An unpaired t-test was used to compare the mean between groups. All graphs related to phenotype analysis were generated using GraphPad Prism 8 software (GraphPad Software, La Jolla, CA). Graphs displayed the mean ± standard deviation (SD). All graphs related to RNA sequencing analysis of mouse muscle tissues were performed using R software version 4.0.4. Statistical analyses were performed using SPSS (version 19.0; IBM-SPSS, Chicago, IL). Two-sided $p < 0.05$ was considered to be statistically significant.

## Acknowledgements

The authors would like to express their gratitude to Dr. Ching H Wang for his critical reading and editing of this manuscript and Dr. Y Zhu for his assistance in the analysis of the RNA sequencing data. This study received support from the following grants: National High Level Hospital Clinical Research Funding (High Quality Clinical Research Project of Peking University First Hospital) (No. 2022CR69 to HX), National Natural Science Foundation of China (No. 82171393 to HX), Natural Science Foundation of Beijing Municipality (No. 7212116 to HX), National Key Research and Development Program of China (No. 2016YFC0901505 to HX), Beijing Key Laboratory of Molecular Diagnosis and Study on Pediatric Genetic Diseases (No. BZ0317 to HX), Research Foundation for Youth Talents of the First Affiliated Hospital of Nanchang University (No. YFYPY202223 to DT), and Natural Science Foundation of Beijing Municipality (No. 7242149 to HL).

# Additional information

## Funding

| Funder | Grant reference number | Author |
| --- | --- | --- |
| National High Level Hospital Clinical Research Funding | 2022CR69 | Hui Xiong |
| National Natural Science Foundation of China | 82171393 | Hui Xiong |
| Natural Science Foundation of Beijing Municipality | 7212116 | Hui Xiong |
| National Key Research and Development Program of China | 2016YFC0901505 | Hui Xiong |
| Beijing Key Laboratory of Molecular Diagnosis and Study on Pediatric Genetic Disease | BZ0317 | Hui Xiong |
| Research Fundation for Youth Talents of the First Affiliated Hospital of Nanchang University | YFYPY202223 | Dandan Tan |
| Natural Science Foundation of Beijing Municipality | 7242149 | Huaxia Luo |

The funders had no role in study design, data collection and interpretation, or the decision to submit the work for publication.

## Author contributions

Dandan Tan, Huaxia Luo, Funding acquisition, Investigation, Methodology, Writing – original draft, Writing – review and editing; Yidan Liu, Luzheng Xu, Jieyu Liu, Investigation, Methodology; Qiang Shen, Methodology; Xingbo Long, Methodology, Writing – original draft; Nanbert A Zhong, Conceptualization, Methodology, Writing – original draft, Writing – review and editing; Hong Zhang, Investigation, Methodology, Writing – original draft, Project administration, Writing – review and editing; Hui Xiong, Conceptualization, Funding acquisition, Investigation, Methodology, Writing – original draft, Project administration, Writing – review and editing

## Author ORCIDs

Dandan Tan  https://orcid.org/0000-0002-1899-5988
Hui Xiong  https://orcid.org/0000-0003-4138-2992

## Ethics

All procedures were approved by the Animal Ethics Committee of Peking University First Hospital (J202027) and followed the guidelines of the Care and Use of Laboratory Animals.

Reviewer #1 (Public review): https://doi.org/10.7554/eLife.94288.4.sa1
Author response https://doi.org/10.7554/eLife.94288.4.sa2

---

# Additional files

## Supplementary files

Supplementary file 1. Differentially expressed genes in all cell clusters in scRNA-seq analysis.

Supplementary file 2. Differentially expressed genes (DEGs) (1136 upregulated and 884 downregulated) at least twofold (p<0.05) in dy$^H$/dy$^H$ (KO) muscle samples relative to the wild-type (WT) muscle samples.

Supplementary file 3. Comparison between the dy$^H$/dy$^H$ mouse with other *Lama2*-deficient mice.

Supplementary file 4. Primers of PCR amplifications for Genotype identification.

Supplementary file 5. MRI examination imaging sequences.

MDAR checklist

## Data availability

The datasets generated during this study are available from Dryad website at https://doi.org/10.5061/dryad.3j9kd51xs.

The following dataset was generated:

| Author(s) | Year | Dataset title | Dataset URL | Database and Identifier |
|---|---|---|---|---|
| Tan D, Liu Y, Luo H, Shen Q, Long X, Xu L, Liu J, Zhong N, Zhang H, Xiong H | 2025 | A novel mouse Model for lama2-related muscular dystrophy with analysis of molecular pathogenesis and clinical Phenotype | https://doi.org/10.5061/dryad.3j9kd51xs | Dryad Digital Repository, 10.5061/dryad.3j9kd51xs |

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
