## [Editor Report · eLife Assessment]

This **useful** manuscript reports on a new mouse model for LAMA2-MD, a rare but very severe congenital muscular dystrophy. The knockout mice were generated by removing exon3 in the Lama2 gene, which results in a frameshift in exon4 and a premature stop codon. These animals lack any laminin-alpha2 protein and confirm results from previous Lama2 knockout models. Additionally, this study includes weak transcriptomics data that might be a good resource for the field. However, experimental evidence, methods, and data analyses supporting the main claims of the manuscript are **incomplete**.

---

## [Referee Report · Reviewer #1 (Public review)]

Strengths:

This work adds another mouse model for LAMA2-MD that re-iterates the phenotype of previously published models. Such as dy3K/dy3K; dy/dy and dyW/dyW mice. The phenotype is fully consistent with the data from others.

One of the major weaknesses of the manuscript initially submitted was the overinterpretation and the overstatements. The revised version is clearly improved as the authors toned-down their interpretation and now also cite the relevant literature of previous work.

Comments on revisions:

This is the second revision of a paper focusing on the generation of a CRISPR/Cas9-engineered mouse model for LAMA2-MD. I have reviewed the initial submission, the first revision, and now this second revision. While there have been improvements, several issues still need to be addressed by the authors. I will outline these points without dividing them into major and minor categories:

Introduction:

The statement regarding existing mouse models requires correction: The claim, "They were established in the pre-gene therapy era, leaving trace of engineering, such as bacterial elements in the Lama2 gene locus, thus unsuitable for testing various gene therapy strategies," is inaccurate. Current mouse models can indeed be used for testing gene therapy strategies, regardless of whether they contain elements in the Lama2 locus. The primary consideration is whether or not they express laminin-alpha2. Please revise this statement.

Results Section:

scRNA-seq:

The authors note that they analyzed "a total of 8,111 cells from the dyH/dyH mouse brain and 8,127 cells from the WT mouse brain were captured using the 10X Genomics platform (Figure supplement 4A, B)." This is too few cells to support firm conclusions. Furthermore, there is a discrepancy in the referred figure S4, which indicates that 10,094 cells were analyzed for dyH/dyH mice and 10,496 for wild-type mice. Please correct this inconsistency.

Figure 5C displays differences in cell populations between wild-type and dyH/dyH mice. Given the low number of cells analyzed and the lack of replicates, these differences cannot be considered reliable. More samples should be analyzed to support these findings.

The data suggest a defect in the BBB for dyH/dyH mice, but this conclusion is based on minimal cell counts and remains purely correlative. If BBB issues exist, experimental validation is necessary, such as injecting dyes into the bloodstream to detect any leakage. I have previously highlighted this in my comments on earlier manuscript versions.

Bulk RNA-seq:

The number of samples analyzed here is substantial, making the data potentially more robust. These data could serve as a valuable resource for other researchers. However, it is important to note that all data are correlative and do not provide functional insights.

Overall:

The manuscript still lacks significant insights, partly because existing mouse models for LAMA2-MD have been extensively analyzed. While the bulk RNA-seq data offer some value as a resource, I recommend that the authors re-assess their writing and further temper their interpretations of the findings.

---

## [Author Response]

The following is the authors’ response to the previous reviews

**Reviewer #1 (Public Review):**
This work adds another mouse model for *LAMA2*-MD that re-iterates the phenotype of previously published models. Such as dy3K/dy3K; dy/dy and dyW/dyW mice. The phenotype is fully consistent with the data from others.

Thank you for the valuable comments and good suggestions you have proposed, and we have added information and analysis of another mouse model for *LAMA2*-MD in the updated version 2 of this manuscript.

One of the major weaknesses of the manuscript initially submitted was the overinterpretation and the overstatements. The revised version is clearly improved as the authors toned-down their interpretation and now also cite the relevant literature of previous work.

Thank you for the good comments you have proposed, and we have carefully corrected the overinterpretation and overstatements in the previous updated version.

Unfortunately, the data on RNA-seq and scRNA-seq are still rather weak. scRNA-seq was conducted with only one mouse resulting in only 8000 nuclei. I am not convinced that the data allow us to interpret them to the extent of the authors. Similar to the first version, the authors infer function by examining expression. Although they are a bit more cautious, they still argue that the BBB is not functional in dy^H^/dy^H^ mice without showing leakiness. Such experiments can be done using dyes, such as Evans-blue or Cadaverin. Hence, I would suggest that they formulate the text still more carefully.

Thank you for the valuable suggestions. We also agree that we should perform more related functional experiments such as Evans-blue or Cadaverin to confirm the impaired BBB. However, the related functional experiments haven’t been done due to the first author has been working in clinic. While, we have added the "Limitations" part, and made statements in the Limitations part with "Even though RNA-seq and scRNA-seq have been performed, the data of scRNA-seq are still insufficient due to the limited number of mouse brains. This study has provided potentially important information for the molecular pathogenetic mechanisms of muscular dystrophy and brain dysfunction for *LAMA2*-CMD, however, some related functional experiments have not been further performed".

A similar lack of evidence is true for the suggested cobblestone-like lissencephaly of the mice. There is no strong evidence that this is indeed occurring in the mice (might also be a problem because mice die early). Hence, the conclusions need to be formulated in such a way that readers understand that these are interpretations and not facts.

Thank you for the valuable suggestions. We do agree with this comment, and have made statement in the Limitations with "This study has provided potentially important information for the molecular pathogenetic mechanisms of muscular dystrophy and brain dysfunction for *LAMA2*-CMD, however, some related functional experiments have not been further performed". Also, for the cobblestone-like lissencephaly which was showed in *LAMA2*-CMD patients while not found in the mouse model, we have added the discussion as "Though the cortical malformations were not found in the dy H/dy H brains by MRI analysis probably due to the small volume in within 1 month old, Thus, the changes in transcriptomes and protein levels provided potentially useful data for the hypothesis of the impaired gliovascular basal lamina of the BBB, which might be associated with occipital pachygyria in LAMA2-CMD patients."

Finally, I am surprised that the only improvement in the main figures is the Western blot for laminin-alpha2. The histology of skeletal muscle still looks rather poor. I do not know what the problems are but suggest that the authors try to make sections from fresh-frozen tissue. I anticipate that the mice were eventually perfused with PFA before muscles were isolated. This often results in the big gaps in the sections.

Thank you for the valuable suggestions. We do agree with this comment and we should make sections from fresh-frozen tissue. Therefore, we have made statement in the Limitations with "Moreover, due to making sections with PFA before muscles isolated, and not from fresh-frozen tissue, there have been big gaps in the sections which do affect the histology of skeletal muscle to some extent."

Overall, the work is improved but still would need additional experiments to make it really an important addition to the literature in the *LAMA*-MD field.

Thank you for all your good comments and the valuable suggestions.

**Reviewer #2 (Public Review):**
This revised manuscript describes the production of a mouse model for *LAMA2*- Related Muscular Dystrophy. The authors investigate changes in transcripts within the brain and blood barrier. The authors also investigate changes in the transcriptome associated with the muscle cytoskeleton. Strengths: (1) The authors produced a mouse model of *LAMA2*-CMD using CRISPR-Cas9. (2) The authors identify cellular changes that disrupted the blood-brain barrier.

Thank you for your good comments.

Weaknesses:The authors throughout the manuscript overstate "discoveries" which have been previously described, published and not appropriately cited.

Thank you for your great suggestion. We have toned-down the interpretations and overstatements throughout the manuscript, and added words such as "potentially", "possible", "some potential clues", "was speculated to probably", and so on.

Alternations in the blood brain barrier and in the muscle cell cytoskeleton in *LAMA2*-CMD have been extensively studied and published in the literature and are not cited appropriately.

Thank you for your great suggestion. We do agree with that alternations in the muscle cell cytoskeleton in *LAMA2*-CMD have been extensively studied and published, and the related literatures have been cited in the updated version 2.0. However, alternations in the blood brain barrier in *LAMA2*-CMD haven’t been extensively studied, only some papers (such as PMID: 25392494, PMID: 32792907) have investigated or discussed this issue.

The authors have increased animal number to N=6, but this is still insufficient based on Power analysis results in statistical errors and conclusions that may be incorrect.

Thank you for your great suggestion. We do agree that the animal number should be increased for Power analysis, and we have added statements in the Limitations with "Finally, due to the limited number of animal samples for the Power analysis, the statistical errors and conclusions might be affected."

The use of "novel mouse model" in the manuscript overstates the impact of the study.

Thank you for your great suggestion. We have changed the statement "novel mouse model" throughout the manuscript except the title.

All studies presented are descriptive and do not more to the field except for producing yet another mouse model of *LAMA2*-CMD and is the same as all the others produced.

Thank you for your comment. We do agree that further functional experiments have not been performed to reveal and confirm the pathogenesis. However, the analysis of phenotype was systematic and comprehensive, including survival time, motor function, serum CK, muscle MRI, muscle histopathology in different stages, and brain histopathology. Moreover, RNA-seq and scRNA-seq in *LAMA2*-CMD have been seldom performed before, and the data in this study could provide potentially important information for the molecular pathogenetic mechanisms of muscular dystrophy and brain dysfunction for *LAMA2*-CMD.

Grip strength measurements are considered error prone and do not give an accurate measurement of muscle strength, which is better achieved using ex vivo or in vivo muscle contractility studies.

Thank you for your great suggestion. We do agree that grip strength measurements are considered error prone and do not give an accurate measurement of muscle strength. And we have added related statement in the Limitations with "Grip strength measurements used in this study are considered error prone and do not give an accurate measurement of muscle strength, which would be better achieved using ex vivo or in vivo muscle contractility studies."

A lack of blinded studies as pointed out of the authors is a concern for the scientific rigor of the study.

Thank you for your great suggestion. We performed the studies with those scoring outcome measures not blinded to the groups. Actually, it was very easy to discriminate the dy^H^/dy^H^ groups from the WT/Het mice due to that the dy^H^/dy^H^ mice showed much smaller body shape than other groups from as early as P7 .

**Recommendations for the authors:**

**Reviewer #2 (Recommendations For The Authors):**
There are multiple grammatical errors throughout the manuscript which should be corrected.

Thank you for your recommendation. We have carefully corrected the grammatical errors within the manuscript.

The authors mention no changes in intestinal muscles, but it is unclear if they are referring to skeletal or smooth muscle.

Thank you for your good comment. The intestinal muscles with no changes in this study are referring to smooth muscle, and we have changes the description into intestinal smooth muscles.